# Explicitly Disentangled Representations in Object-Centric Learning

**Riccardo Majellaro**[*]                                    *riccardomajellaro@gmail.com*

**Jonathan Collu**[*]                                        *jonathancollu1998@gmail.com*

**Aske Plaat**[1]                                            *a.plaat@liacs.leidenuniv.nl*

**Thomas M. Moerland**[1]                                    *t.m.moerland@liacs.leidenuniv.nl*

[1]*Leiden Institute of Advanced Computer Science, Leiden University*

**Reviewed on OpenReview:** *https://openreview.net/forum?id=r8UFp9olQO*

## Abstract

Extracting structured representations from raw visual data is an important and long-standing challenge in machine learning. Recently, techniques for unsupervised learning of object-centric representations have raised growing interest. In this context, enhancing the robustness of the latent features can improve the efficiency and effectiveness of the training of downstream tasks. A promising step in this direction is to disentangle the factors that cause variation in the data. Previously, Invariant Slot Attention disentangled position, scale, and orientation from the remaining features. Extending this approach, we focus on separating the *shape* and *texture* components. In particular, we propose a novel architecture that biases object-centric models toward disentangling shape and texture components into two non-overlapping subsets of the latent space dimensions. These subsets are known *a priori*, hence before the training process. Experiments on a range of object-centric benchmarks reveal that our approach achieves the desired disentanglement while also numerically improving baseline performance in most cases. In addition, we show that our method can generate novel textures for a specific object or transfer textures between objects with distinct shapes.

## 1  Introduction

A key challenge in deep learning is to learn data representations that can be leveraged in a multitude of downstream tasks. The promise of exploiting large unlabeled datasets has driven research in unsupervised and self-supervised settings, despite the inherent complexity of this task. In the context of computer vision, recent works focus on representing, without supervision, visual scenes composed of multiple objects as sets of latent vectors centered around objects (Burgess et al., 2019; Greff et al., 2019; Engelcke et al., 2019; 2021; Locatello et al., 2020). This object-centric strategy allows for more structured representations compared to single flat vector encodings of complete images. With single flat vectors, multiple objects can be entangled in components of the same representation or separated into distinct dimensions. However, even when separated, objects still cannot share features of common properties such as their position. Instead, with object-centric vectors, visual entities are encoded as individual vectors within the same space, improving generalization and interpretability as entities are clearly divided but described by the same features. On top of that, these representations naturally scale to an increasing number of objects. Our work is based on this approach, often referred to as object-centric representation learning.

Disentanglement is considered a central aspect of learning robust representations of complex and structured visual scenes (Bengio et al., 2013). The term refers to the encoding of different factors of variation in the data

---

[*]Work done while at LIACS, Leiden University.

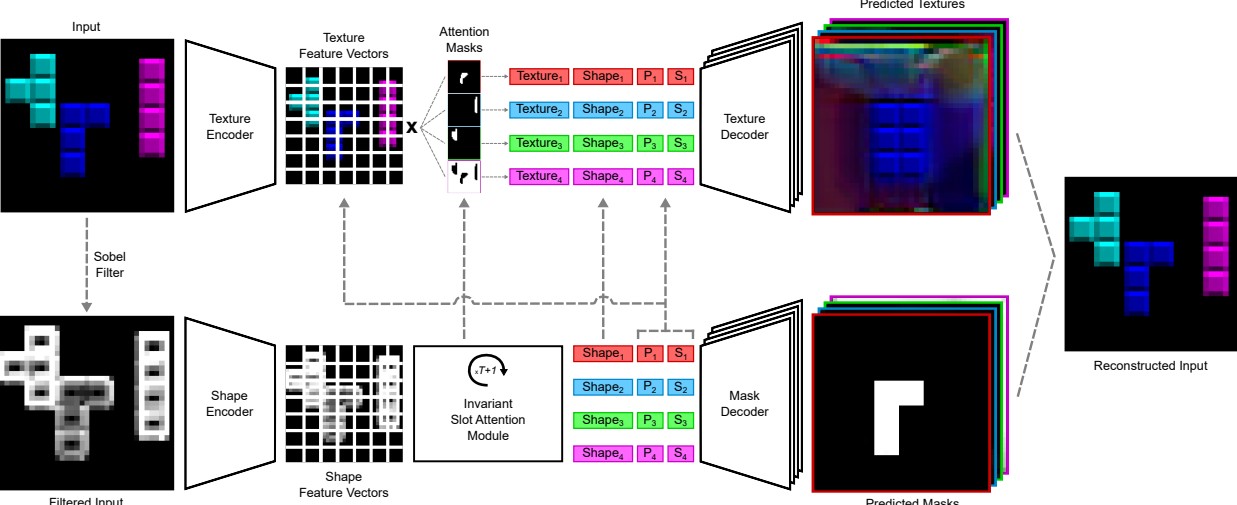

Figure 1: Architecture of DISA. The input image (top-left) is first fed through a Sobel filter to partly remove texture information (bottom-left). This filtered image is then encoded and passed through an Invariant Slot Attention (Biza et al., 2023) module to extract object-centric shape ($\text{Shape}_i$), position ($\text{P}_i$), and scale ($\text{S}_i$) vectors (bottom-middle). This information is used to decode a mask for each object (bottom-right). The initial input image is encoded (top-left) and then combined with the object attention masks from the ISA module to produce a texture vector ($\text{Texture}_i$) per object (top-middle). These texture representations are combined with their associated shape, position, and scale to decode the textures of the objects (top-right). Note that we need to add the shape, location, and scale information as the texture decoding should fit the already predicted masks. Finally, the decoded masks and textures are combined into an image reconstruction (right).

into separate dimensions of the latent space. Disentanglement can provide different advantages, including (1) enhanced interpretability of the learned features, (2) compositional generalization beyond the training data distribution, and (3) helping the learning of relevant yet unknown downstream tasks (e.g., in the context of reinforcement learning). Prior probabilistic work on object-centric representation learning (Burgess et al., 2019; Greff et al., 2019) exploited VAEs (Kingma & Welling, 2013) and spatial broadcast decoders (Watters et al., 2019), which helped the disentanglement of the features. In contrast, the non-probabilistic approach of Slot Attention (SA) (Locatello et al., 2020) extracts object vectors based on a competitive cross-attention mechanism. However, the latter method lacks clear property-level disentanglement, as shown in Singh et al. (2022). Biza et al. (2023) proposed an extension of SA, namely Invariant Slot Attention (ISA), aimed at learning object representations invariant to position, orientation, and scale, allowing for the explicit disentanglement of those three factors. However, to the best of our knowledge, no research has been carried out on the explicit disentanglement of the *texture* and *shape* dimensions in unsupervised object-centric learning, especially with non-probabilistic models. Note that, by the term explicit, we refer to the process of choosing which latent components will be responsible for encoding certain factors, knowing them *a priori*. In contrast, probabilistic models such as VAE or $\beta$-VAE (Higgins et al., 2016) cannot enforce which properties to disentangle and control where they should be encoded. Possible benefits of explicit disentanglement include easier interpretation and more reliable modeling of structured latent spaces. Moreover, it creates the possibility of designing objectives that exploit this prior knowledge from the beginning, removing the need to analyze and interpret the latent components during or after the training process.

Our work focuses on biasing object-centric models toward explicitly disentangling the groups of features responsible for encoding shape and texture information into two non-overlapping subsets of the latent space dimensions. By building upon ISA, we combine the separation of position and scale with that of texture shape. As these are core properties of objects, this result can be crucial for further improving the internal structure of the learned representations, especially in a non-probabilistic setting. To disentangle texture

and shape, we design a novel architecture that employs two encoder-decoder pairs. The first pair encodes shape information and decodes object masks; the second pair represents and predicts object textures. In addition, a filter (e.g. Sobel filter, Sobel & Feldman (1968)) is used to partly remove texture information from the input image before feeding it to the shape encoder, helping to prevent texture information from flowing into shape-related latent components. Furthermore, a regularization term is introduced to reduce the variance of the latent features across slots, helping to deter shape information from being included in texture-related components and vice versa. We name our approach DIsentangled Slot Attention (DISA). DISA is characterized by a latent space divided into four distinct groups known since the beginning of the training process: the first two encoding texture and shape, the third position, and the last one scale (Figure 1).

Our experimental evaluations indicate that, in most cases, DISA introduces the desired disentanglement property, enabling novel generative and compositional capabilities. At the same time, DISA is competitive with or outperforms the baselines at scene decomposition on three well-known synthetic datasets, while achieving significantly improved reconstruction quality. As such, DISA enhances both the interpretability and reconstruction quality (information content) of learned object-centric image representations.

## 2 Background

### 2.1 Slot Attention

The *Slot Attention* (SA) module is an architectural component proposed by Locatello et al. (2020) based on the concept of cross-attention (Luong et al., 2015; Vaswani et al., 2017). Its aim is to bind a set of latent vectors called *slots* to different objects, seen as parts of the input perceptual representations, through multiple iterations of a competitive attention mechanism. The set of $N_s$ slots $S \in \mathbb{R}^{N_s \times d_s}$ is initialized either as learnable vectors or by independently sampling each from a Gaussian distribution with shared and learnable parameters. The $N_x$ input representations are obtained by augmenting the feature vectors $X \in \mathbb{R}^{N_x \times d_x}$ at the output of a CNN backbone with positional embeddings $g(G)$, where $G$ is a flattened grid encoding absolute positions and $g$ a linear mapping from the grid coordinates space to dimension $d_x$. These augmented inputs are then passed through an MLP $f$, normalized using layer normalization (LayerNorm) (Ba et al., 2016) and finally projected as keys and values $K, V \in \mathbb{R}^{N_x \times d}$ of a common dimension $d$ through the linear transformations $k, v : \mathbb{R}^{d_x} \to \mathbb{R}^d$. At each of the $T$ iterations, the slots from the previous round are (independently) normalized, mapped as queries $Q \in \mathbb{R}^{N_s \times d}$ by $q : \mathbb{R}^{d_s} \to \mathbb{R}^d$, and then refined.

At a given refinement step $t$ of the Slot Attention mechanism, the previous slots $S^{t-1}$ are mapped as queries and fed to the competitive cross-attention module, which yields one vector $U_i$ for each $Q_i$, referred to as *update*. The updates are then passed as inputs to a shared Gated Recurrent Unit (GRU) (Cho et al., 2014), while the associated slot vectors from $S^{t-1}$ are set as hidden states. The outputs, which are updated hidden states, are transformed by a shared MLP with a residual connection (He et al., 2016), resulting in the new slot vectors $S^t$. The competition between slots for explaining part of the input is induced by an additional normalization in the cross-attention mechanism, performed by using the softmax function over the queries before linearly normalizing over the keys. Formally, the attention coefficients are computed as

$$\hat{w}_{ij} = \frac{e^{M_{ij}}}{\sum_{l=1}^{N_s} e^{M_{lj}}} \quad \text{where} \quad M = \frac{QK^\top}{\sqrt{d}} \in \mathbb{R}^{N_s \times N_x} \ , \tag{1}$$

producing a probability distribution over the slots for each input vector. After that, the update vectors $U_i$ are calculated as

$$U_i = \sum_{j=1}^{N_x} w_{ij} V_j \quad \text{with} \quad w_{ij} = \frac{\hat{w}_{ij}}{\sum_{l=1}^{N_x} \hat{w}_{il}} \quad \text{such that} \quad \sum_{j=1}^{N_x} w_{ij} = 1 \ . \tag{2}$$

It is possible to exploit Slot Attention, coupled with a backbone, as a structured encoder in an autoencoder architecture. The model is trained end-to-end to represent an input image as a set of latent vectors that are decoded back to the original image. A shared spatial broadcast decoder (Watters et al., 2019) is used

to decode the slots individually: each is repeated in a $H \times W$ grid, augmented with positional embeddings, and passed through a series of convolutional layers. The output is an $H \times W \times 4$ tensor, where the first three channels correspond to RGB components, while the last one is the alpha mask of the decoded object. The reconstructed image is computed as the sum of the RGB predictions weighted by the alpha masks. At training time, the model parameters are optimized to minimize the mean squared error (MSE) between the reconstructed and input pixels.

## 2.2  Slot-Centric Reference Frames

With *Invariant Slot Attention* (ISA), Biza et al. (2023) presented a mechanism for introducing per-object invariances to introduce transformations in the latent representations. To achieve this, slot-centric relative grids are produced by translating, scaling, and rotating the absolute positional encodings, with the aim of processing the feature vectors in canonical reference frames. The rotation invariance is not covered in this section as it is not employed in our method.

At the first iteration of Slot Attention, translation and scaling factors $S_i^{\text{pos}}$ and $S_i^{\text{scale}}$ associated with the slot $S_i$ can be randomly sampled or learned. The relative grid is then computed and its projection is summed to the feature vectors X to obtain per-object keys and values $K^i, V^i \in \mathbb{R}^{N_x \times d}$:

$$G^i = \frac{G - S_i^{\text{pos}}}{S_i^{\text{scale}}} \quad \text{and} \quad K^i = f(k(X) + g(G^i)) \quad , \quad V^i = f(v(X) + g(G^i)) \,. \tag{3}$$

$K^i$ is then processed by competitive cross-attention (Equation 1) with the query derived from $S_i$ to yield $M_i$, finally used to compute $w_i$ and $U_i$ with $V^i$ (Equation 2). During the next iteration, $w_i$ is used to extract the new translation and scaling factors as

$$S_i^{\text{pos}} = \frac{\sum_{j=1}^{N_x} w_{ij} G_j}{\sum_{j=1}^{N_x} w_{ij}} \quad \text{and} \quad S_i^{\text{scale}} = \sqrt{\frac{\sum_{j=1}^{N_x} (w_{ij} + \epsilon)(G_j - S_i^{\text{pos}})^2}{\sum_{j=1}^{N_x} (w_{ij} + \epsilon)}} \,, \tag{4}$$

which are again employed to calculate the relative grid of the $i$-th object. Note that the grids have two dimensions (horizontal and vertical) with values in the range $[-1, 1]$.

During the decoding phase, the relative frames are used again to provide back position and scale information and ensure invariant decoding of the objects. In this case, the last translation and scaling factors are exploited to compute the grids as in Equation 3 and added to the broadcasted slots.

## 3  Related Work

**Object-centric representation learning**  The method we propose is aimed at models learning object-centric representations of visual scenes. These approaches, learning in an unsupervised or self-supervised manner, are capturing increasingly more attention. A first line of work in this direction is defined by AIR (Eslami et al., 2016). AIR performs probabilistic inference employing a recurrent neural network (RNN) that attends and processes one object in a scene at a time. Subsequent work extended this idea in order to solve some of its limitations, such as the SQAIR approach (Kosiorek et al., 2018), SuPAIR (Stelzner et al., 2019), and others (Crawford & Pineau, 2019; Jiang et al., 2019; Lin et al., 2020). Further advances in object-centric representation learning include Tagger (Greff et al., 2016), Multi-Entity Variational Autoencoder (MVAE) (Nash et al., 2017), and Neural Expectation Maximization (N-EM) (Greff et al., 2017) with its extension R-NEM (Van Steenkiste et al., 2018). More recently, Burgess et al. (2019); Greff et al. (2019); Engelcke et al. (2019; 2021) achieved meaningful decomposition of non-trivial scenes with a variable number of objects, e.g., in the CLEVR dataset (Johnson et al., 2017). Finally, Slot Attention (Locatello et al., 2020), along with various extensions (Kipf et al., 2021; Singh et al., 2021; Chang et al., 2022; Jia et al., 2022; Biza et al., 2023; Kori et al., 2023), introduces a non-probabilistic iterative mechanism that is competitive with its predecessors while being faster to train and more memory efficient.

**Disentanglement in object-centric learning**  Probabilistic models such as in Burgess et al. (2019); Greff et al. (2019) can obtain a degree of disentanglement due to their foundation on the VAE framework. Differently yet still in a probabilistic setting, Anciukevicius et al. (2020) works toward the explicit disentanglement of position and depth. Mansouri et al. (2022), instead, exploits weak supervision from sparse perturbations and causal representation learning to disentangle object properties. In a non-probabilistic setting, Singh et al. (2022) learns disentangled representations in a non-explicit manner, while Biza et al. (2023) introduces invariance to changes in position, scale, and rotation with the use of slot-centric reference frames, allowing for the explicit disentanglement of those three factors. We exploit the concept of slot-centric reference frames in our work. Furthermore, similarly to Anciukevicius et al. (2020); Biza et al. (2023), we possess a priori knowledge of which of the latent dimensions are associated with a disentangled factor, in contrast with, e.g., Burgess et al. (2019); Greff et al. (2019); Singh et al. (2022), where this knowledge can only emerge after training. Prabhudesai et al. (2020) explicitly disentangles shape and texture in object-centric learning but relies on ground-truth bounding boxes and multiple viewpoints per static scene (more in Appendix C). To the best of our knowledge, no research dealt with the explicit disentanglement of texture and shape factors in unsupervised object-centric learning, which is the aim of our work.

**Texture and shape disentanglement**  Outside the scope of object-centric learning, approaches including Shu et al. (2018); Lorenz et al. (2019); Yan et al. (2023) attempt to disentangle shape and texture in the domain of single-object images. Deforming Autoencoders (Shu et al., 2018) employs a pair of decoders, of which one synthesizes appearance in a deformation-free coordinate system, while the other estimates a deformation field that warps the texture into the input image. One aspect that aligns with our work is the adoption of two separate decoders for texture and shape synthesis. Lorenz et al. (2019) aims to disentangle appearance and shape in the representations of multiple parts of a single object class, without supervision. To do so, they introduce into the reconstruction task three invariance and equivariance constraints, by exploiting texture and shape transformations of an input image (more in Appendix C). Finally, TSD-GAN (Yan et al., 2023) uses an adversarial framework to learn to both reconstruct the item in an input image and mix it with the item from another sample.

## 4    Methods

The primary aim of our work is to induce object-centric models to learn representations characterized by the explicit, and thus known a priori, separation of the groups of features encoding texture and shape information. To allow for this, we propose an architectural design along with a simple regularization of the latent space. We apply our approach to Invariant Slot Attention (ISA) and call the resulting model DISA, for DIsentangled Slot Attention (Figure 1). The choice of ISA enables us to further integrate the explicit disentanglement of position and scale in our model by leveraging slot-centric reference frames.

### 4.1    Architectural Design

Consider a generic architecture for unsupervised object discovery, encoding an image into a set of latent representations (slots) and decoding them as pairs of mask and texture reconstructions. We want the slots to have a first subset of their components encoding texture information only (e.g. material and color), a second one encoding only shape information, and the rest of the dimensions representing what is still unexplained, such as position and scale. To store shape information within the shape group, we decode object masks exclusively using those components, as knowledge of an entity's shape is essential for accurate masking prediction. Similarly, we want to force texture information inside the texture group, which we achieve through a separate decoder that infers object textures from those components. However, as textures need to be decoded according to the associated object shape, shape information would also flow into the texture group. To avoid this, we let the texture decoder exploit both texture and shape subsets. Nevertheless, the current design (with two decoders) allows the shape components to encode all the necessary information, which would thwart all our efforts. Therefore, we ultimately employ two distinct encoders: one extracts and represents texture information from the input image into the texture group, while the other encodes the shape information inside the shape group starting from a filtered input. The filter should ideally remove all

the texture information from the image, but in practice, it can be sufficient to erase part of it to bias the model as we desire.

As it is, the design does not yet deal with position and scale information, which would thus end up in the texture and shape groups instead of two different ones. Moreover, the texture components could still include some shape-related information and vice versa. We address these limitations in the next section with the implementation of DISA.

## 4.2 Disentangled Slot Attention

In order to implement the design described in Section 4.1 while addressing its stated limitations, we extend Invariant Slot Attention according to the above principles (Figure 1) and train it with the reconstruction loss along with a simple latent space regularization.

Let $\mathcal{X} \in [0,1]^{H \times W \times 3}$ be an RGB input image, $\hat{\mathcal{X}} \in \mathbb{R}_+^{H \times W}$ be $\mathcal{X}$ processed with a Sobel filter, and $\Phi_{\text{tex}} : [0,1]^{H \times W \times 3} \to \mathbb{R}^{N_x \times d_x}$, $\Phi_{\text{shape}} : \mathbb{R}_+^{H \times W} \to \mathbb{R}^{N_x \times d_x}$ the texture and shape CNN backbones. First, the texture and shape input representations are obtained respectively as $\text{X}_{\text{tex}} = \Phi_{\text{tex}}(\mathcal{X})$ and $\text{X}_{\text{shape}} = \Phi_{\text{shape}}(\hat{\mathcal{X}})$. Then, the ISA mechanism (Section 2.2) is applied to $\text{X}_{\text{shape}}$: at each iteration $t$, the keys and values relative to an object $i$ are computed as

$$\text{K}_{\text{shape}}^i = f_{\text{shape}}(k_{\text{shape}}(\text{X}_{\text{shape}}) + g_{\text{shape}}(\text{G}^i)) \quad \text{and} \quad \text{V}_{\text{shape}}^i = f_{\text{shape}}(v_{\text{shape}}(\text{X}_{\text{shape}}) + g_{\text{shape}}(\text{G}^i)) , \quad (5)$$

where $\text{G}^i$ is calculated using $\text{S}_i^{\text{pos}}$ and $\text{S}_i^{\text{scale}}$ from the previous iteration. $\text{K}_{\text{shape}}^i$ and $\text{V}_{\text{shape}}^i$ are used with $\text{Q}_i$ as in Equations 1,2 to yield $w_i$ and $\text{U}_i$, necessary to get the new shape-related vector $\text{S}_i^{\text{shape}} = h_{\text{shape}}(\text{S}_i^{\text{shape}}, \text{U}_i)$. $h_{\text{shape}}$ represents the GRU followed by a residual MLP. At $t = T$, the final shape vectors, which are invariant to position and scale, are returned and employed in an additional iteration $t = T + 1$ to get the final attention coefficient $w_i$, position vector $\text{S}_i^{\text{pos}}$, and scale vector $\text{S}_i^{\text{scale}}$.

As the objects in the image are already discovered, there is no need to apply an additional ISA mechanism to extract the texture-related vectors. Instead, the last $w_i$, $\text{S}_i^{\text{pos}}$, and $\text{S}_i^{\text{scale}}$ can be exploited to combine $\text{X}_{\text{tex}}$ (in a position and scale invariant manner) according to the final ISA iteration as

$$\text{S}_i^{\text{tex}} = h_{\text{tex}} \left( \sum_{j=1}^{N_x} w_{ij} \text{V}_{\text{tex}_j}^i \right) \quad \text{with} \quad \text{V}_{\text{tex}}^i = f_{\text{tex}}(v_{\text{tex}}(\text{X}_{\text{tex}}) + g_{\text{tex}}(\text{G}^i)) . \quad (6)$$

In this case, $h_{\text{tex}}$ is simply an MLP without residual connection and not preceded by a GRU. Note that $\text{S}^{\text{tex}} \in \mathbb{R}^{N_s \times d_{\text{tex}}}$, $\text{S}^{\text{shape}} \in \mathbb{R}^{N_s \times d_{\text{shape}}}$, and $\text{S}^{\text{pos}}, \text{S}^{\text{scale}} \in \mathbb{R}^{N_s \times 2}$, hence each object is represented by a slot of dimension $d_{\text{tex}} + d_{\text{shape}} + 2 + 2$.

Finally, the texture and mask of the $i$-th object are inferred by two spatial broadcast decoders, where the broadcasted slots are augmented with the relative frame obtained with $\text{S}_i^{\text{pos}}$ and $\text{S}_i^{\text{scale}}$. Specifically, the mask decoder input is composed exclusively by the shape vector $\text{S}_i^{\text{shape}}$, while the texture decoder is fed with the concatenation of $\text{S}_i^{\text{tex}}$ and $\text{S}_i^{\text{shape}}$.

We train the parameters of DISA with the reconstruction loss along with a simple regularization on the variance of the latent features across slots:

$$\mathcal{L} = \mathcal{L}_{\text{rec}} + \lambda \left( \frac{1}{d_{\text{tex}}} \sum_{l=1}^{d_{\text{tex}}} \text{Var}(\text{S}^{\text{tex}_l}) + \frac{1}{d_{\text{shape}}} \sum_{l=1}^{d_{\text{shape}}} \text{Var}(\text{S}^{\text{shape}_l}) \right) \quad (7)$$

where $\text{S}^{\text{tex}_l} = (\text{S}_{1l}^{\text{tex}}, \ldots, \text{S}_{N_s l}^{\text{tex}})$, $\text{S}^{\text{shape}_l} = (\text{S}_{1l}^{\text{shape}}, \ldots, \text{S}_{N_s l}^{\text{shape}})$. When the loss is computed over a batch of images, $N_s$ becomes $N_s$ times the number of images in the batch. The second term is introduced with the purpose of removing the remaining shape-related information from $\text{S}^{\text{tex}}$ and texture-related information from $\text{S}^{\text{shape}}$. In fact, the same texture can be encoded slightly differently when belonging to objects with distinct

Table 1: BG-ARI and FG-ARI ($\uparrow$) averaged over 3 seeds and represented as percentages, in the format mean $\pm$ stddev. FG-ARI excludes the background and only considers the foreground objects, while BG-ARI considers both foreground and background. For ISA (on Tetrominoes and CLEVR6) and SA, we perform 10 evaluations of each test image to address the stochasticity arising from the sampling (Appendix D.1).

| | BG-ARI | | | | FG-ARI | | | |
|---|---|---|---|---|---|---|---|---|
| | Tetrominoes | Multi-dSprites | CLEVR6 | CLEVRTex | Tetrominoes | Multi-dSprites | CLEVR6 | CLEVRTex |
| SA | $42.7 \pm 3.2$ | $43.4 \pm 38.5$ | $92.8 \pm 1.3$ | $16.4 \pm 6.9$ | $98.4 \pm 0.5$ | $55.8 \pm 26.0$ | $92.1 \pm 0.4$ | $64.2 \pm 14.6$ |
| ISA | $99.1 \pm 0.5$ | $97.2 \pm 0.9$ | $84.1 \pm 2.0$ | $23.5 \pm 3.6$ | $99.3 \pm 0.3$ | $71.0 \pm 5.5$ | $93.9 \pm 0.4$ | $71.3 \pm 1.8$ |
| DISA | $96.7 \pm 2.0$ | $98.9 \pm 0.1$ | $96.3 \pm 0.1$ | $32.2 \pm 10.0$ | $94.4 \pm 3.6$ | $90.5 \pm 1.0$ | $97.0 \pm 0.2$ | $79.2 \pm 0.2$ |

Table 2: MSE ($\downarrow$) averaged over 3 seeds and represented in the format mean $\pm$ stddev. All the numbers in the table are scaled by $10^4$. For ISA (on Tetrominoes and CLEVR6) and SA, we perform 10 evaluations of each test image to address the stochasticity arising from the sampling (Appendix D.1). DISA significantly surpasses the baselines in terms of reconstruction quality in most cases.

| | Tetrominoes | Multi-dSprites | CLEVR6 | CLEVRTex |
|---|---|---|---|---|
| SA | $6.87 \pm 1.99$ | $22.90 \pm 19.11$ | $5.90 \pm 0.50$ | $60.90 \pm 1.92$ |
| ISA | $6.30 \pm 3.33$ | $4.57 \pm 1.88$ | $6.03 \pm 0.26$ | $37.70 \pm 0.78$ |
| DISA | $2.67 \pm 0.97$ | $2.20 \pm 0.08$ | $2.10 \pm 0.08$ | $41.37 \pm 1.24$ |

shapes, and conversely, an identical shape can be encoded slightly dissimilarly when belonging to objects with unequal textures. By bringing a set of texture (or shape) representations close together we try to minimize these undesired differences as much as possible. At the same time, the reconstruction loss prevents vectors belonging to different textures (or shapes) from collapsing to the same values since, in that case, it would not be possible to accurately differentiate them during the decoding process.

## 5 Experiments

In this section, we evaluate DISA on four well-known multi-object synthetic datasets (Kabra et al., 2019; Karazija et al., 2021): Tetrominoes, Multi-dSprites, CLEVR, and CLEVRTex (Appendix B). For CLEVR, we train on a filtered version of this dataset called CLEVR6. Each training configuration is run three times with different random seeds to account for the stochastic nature of the experiments. Initially, we compare DISA against the baselines (SA and ISA) to assess its ability to reconstruct images and decompose them into objects without supervision. Subsequently, we try to demonstrate that DISA achieves the desired disentanglement property in its latent space. Finally, we show (through qualitative experiments) compositional and generative capabilities derived from the disentangled representations. Note that this work seeks to achieve the desired disentanglement within the latent space of DISA rather than focusing on obtaining state-of-the-art results in unsupervised object discovery and reconstruction quality. Additional experiments are included in Appendix F. The code is available at `https://github.com/riccardomajellaro/disentangled-slot-attention`.

### 5.1 Object Discovery and Reconstruction

This evaluation investigates whether DISA is competitive with the baselines at unsupervised object discovery and image reconstruction. For object discovery, we compare the predicted object masks with the ground truth through the Adjusted Rand Index (ARI) score (Rand, 1971; Hubert & Arabie, 1985) computed both including (BG-ARI) and excluding (FG-ARI) the background masks. The use of the ARI score is in line with Locatello et al. (2020); Biza et al. (2023). As for the reconstruction quality, we employ the mean squared error (MSE). Table 1 summarizes the ARI scores, while Table 2 the MSE. We find that on the datasets that were used, our models are comparable and, in most cases, even outperform both baselines at decomposing

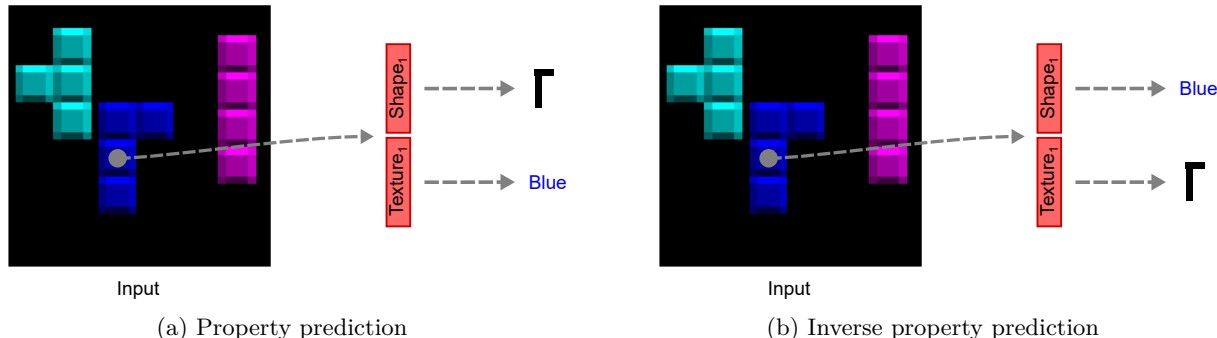

(a) Property prediction

(b) Inverse property prediction

Figure 2: Illustration of the two property prediction tasks that we propose to quantitatively study the degree of texture and shape disentanglement in the representations of DISA. (**a**) Prediction of an object property based on the associated components. For instance, the color from the texture-related latent features. (**a**) Inverse property prediction task, where properties are predicted based on the "wrong" part of the object representation, e.g., the color from the shape-related components.

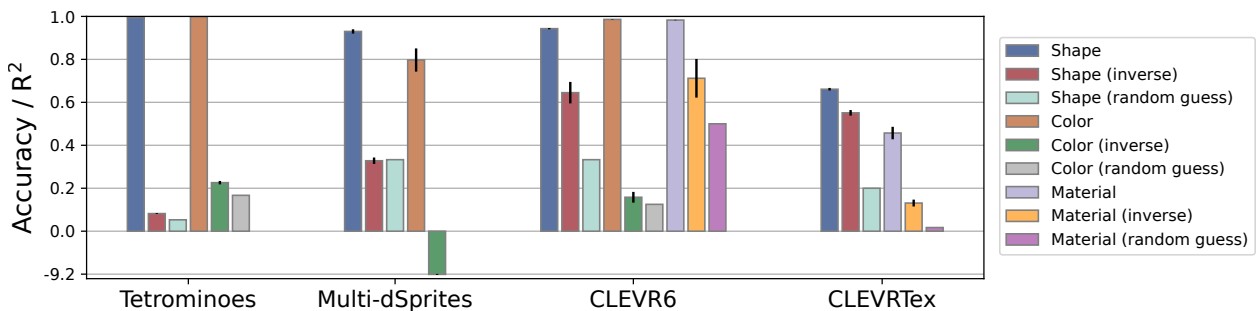

Figure 3: Quantitative results of DISA on the regular and inverse property prediction tasks. The predicted properties are shape, color, and material, which are all categorical variables except for the color on Multi-dSprites. With categorical variables, the prediction accuracy is employed and compared with a baseline random guess. With the numerical one, the $R^2$ score is shown. We report mean and stddev over 3 seeds. On Tetrominoes and Multi-dSprites, DISA correctly encodes texture and shape information into two non-overlapping subsets of its latent space dimensions. On CLEVR6 and CLEVRTex, part of the shape and texture information leaks into incorrect components.

the images into objects. Looking at the MSE, DISA achieves better reconstruction quality than SA and ISA on all the experimented datasets by a considerable margin (except on CLEVRTex, where it performs slightly worse than ISA). Overall, despite not being the primary goal of this research, our models are competitive with and, in most cases, even exceed the performance of the baselines.

## 5.2   Disentanglement Analysis

To verify how effective DISA is in constraining the texture and shape information into the desired components, we utilize property prediction (Figure 2): we attempt to predict the ground-truth property of an object from the subset of its slot where the related information should be present, and also from a subset where the information should not be encoded. To do so, we first train multiple MLPs, each predicting one property of the objects based on the respective part of the slots, such as the shape of a tetromino given its shape vector (Figure 2a). In this task, we expect excellent results as we infer some property leveraging the part of the representations that should encode it. Instead, if we switched the parts and trained another set of MLPs to predict, e.g., the color of an object from its shape-related components, we would ideally expect to obtain

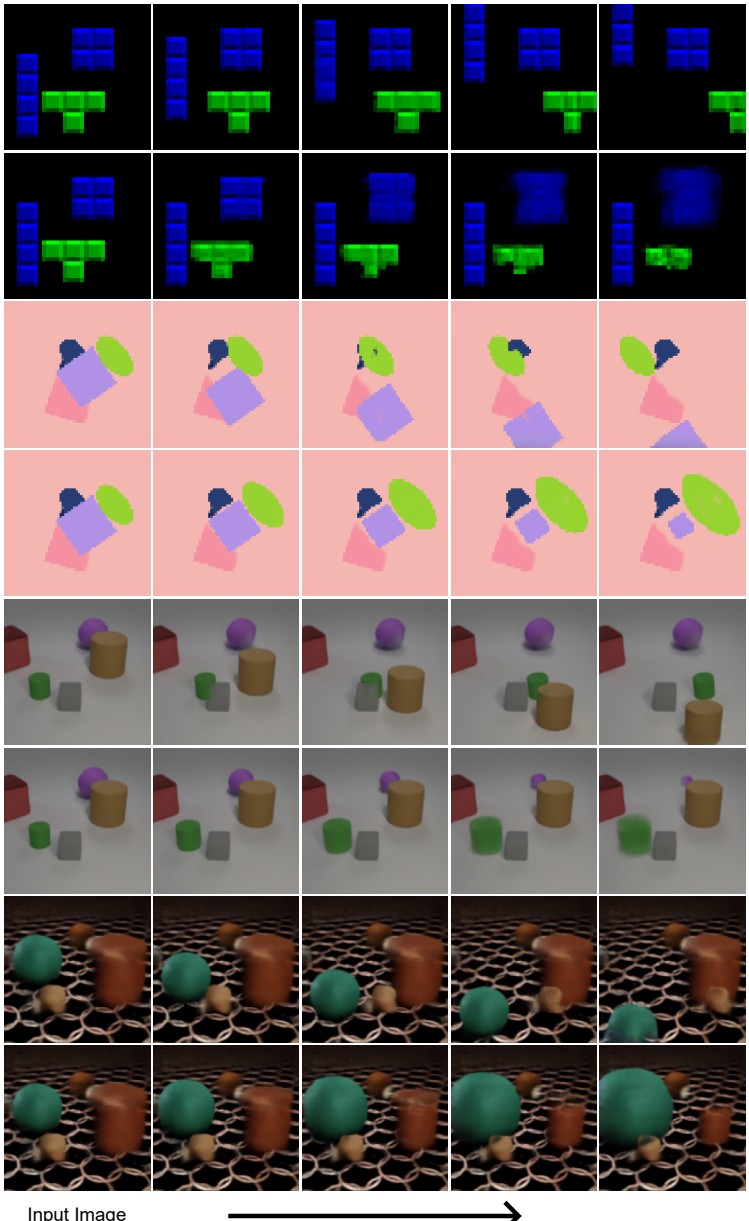

Input Image

Figure 4: Qualitative results showing the position and scale disentanglement on DISA. Given an input image (first column), we select two objects and modify their scale/position factors incrementally over 4 steps (second to fifth columns). Texture and shape are consistent across the steps, while position and scale can be independently varied correctly, suggesting that the models can confine position and scale information in their respective components. The visible limitations are displayed by ISA as well, e.g., the missing parts of an object after moving another object covering it or even the inconsistent masks produced when incrementing the scale over a certain point. Additionally, on Tetrominoes, scaling an object does not yield completely wrong results, which is not trivial as the model only sees shapes of fixed size during training.

poor results. An "inverse" property prediction task is carried out to gain insight into this aspect (Figure 2b). In line with previous literature (Greff et al., 2019; Dittadi et al., 2021; Papa et al., 2022), we measure performance using the accuracy for categorical variables and the coefficient of determination ($R^2$ score) for numerical variables (i.e. only the color in Multi-dSprites). We report the mean and standard deviation

of accuracy/$R^2$ computed over the three seeds on the test set. As baselines, we provide the accuracy of a random guess for each property on every dataset. Moreover, since the objects can be represented in any order, the ground truth labels must be matched with the proper predicted slots. We do so by computing cosine similarities between the target and predicted masks and then coupling each ground truth label with the output of its most similar slot.

The results of these experiments on Tetrominoes, Multi-dSprites, CLEVR6, and CLEVRTex are reported in Figure 3. As expected, DISA reaches near-perfect property prediction scores on all datasets (except for CLEVRTex) when exploiting the correct groups of features. When inverting them, we notice on Tetrominoes and Multi-dSprites that the accuracy of the categorical predictions drops approximately to that of a random guess, while the $R^2$ of the numerical becomes negative. The negative $R^2$ score indicates that the inverse color predictions are worse than those of a model constantly inferring the mean target value. Therefore, these results show that, on the first two datasets, DISA is able to correctly encode texture and shape information into two non-overlapping subsets of its latent space dimensions, achieving the purpose of this work. In the case of CLEVR6, we find that although the color can be successfully restricted within the texture components, part of the shape and material information leaks respectively into the texture and shape latent features. On CLEVRTex, as both ISA and DISA are unable to precisely segment and reconstruct the objects, it is likely that the representations do not contain the necessary information to allow near-perfect accuracy even on the regular property prediction task. Even in this case, part of the shape and texture information leaks into incorrect components.

We further present in Figure 4 qualitative results showing that the position and scale disentanglement properties of DISA are preserved. Given an input image (first column), we select two objects and modify their scale or position factors incrementally over 4 steps (second to fifth columns). As visible, texture and shape are consistent across the steps, while position and scale can be independently varied with success. This suggests that the models are able to confine position and scale information in their respective components. Note that the visible limitations are displayed by ISA as well. For example, the missing parts of an object after moving another object covering it, or even the inconsistent masks produced when incrementing the scale over a certain point. Moreover, it is interesting to see that, on Tetrominoes, scaling an object does not yield completely wrong results. This is not trivial as, for a given shape, only a single variation of its scale is present in the dataset, meaning that the model only sees shapes of fixed size during training.

### 5.3 Compositional and Generative Capabilities

Finally, we also explore the compositional and generative capabilities of DISA derived from the disentanglement of texture and shape through simple qualitative experiments. Precisely, as a first investigation, for a given image from the test set we take the representations of the objects, randomly permute their texture parts, and decode the permuted representations. Ideally, the model would output a reconstruction where the shapes, positions, and sizes of the entities are preserved, while their textures are interchanged and adapted to fit the same masks. In the second experiment, we analyze whether this ability extends to the generation of new objects by providing new sampled textures or shapes to the objects in a scene. To do so, we average the texture (or shape) vectors encoded from a given image and sample from a Gaussian distribution centered on that mean vector. For additional results, including those on Multi-dSprites and CLEVRTex, please see Appendix F.2 and F.3.

Figure 5 reports the compositional results on Tetrominoes and CLEVR6. On Tetrominoes (Figure 5a), transferring the texture components of one object to another translates into passing the color. As visible, the colors are in fact interchanged between objects without affecting the original shapes and positions, coherently with the quantitative results from Figure 3. On CLEVR6 (Figure 5b), colors and materials are correctly transferred between objects while being shaped, scaled, and positioned according to the shape, position, and scale information. Differently from Tetrominoes, this result only partly supports those from Figure 3. In fact, although we expected the colors to interchange accurately, we would have predicted less precise transferring of shapes and materials. However, looking at more samples and, as the high standard deviation in the inverse material prediction suggests, different seeds may be necessary for coherence with the quantitative results. Overall, Figure 5 shows strong compositional generalization, enabling reliable transferring of textures between

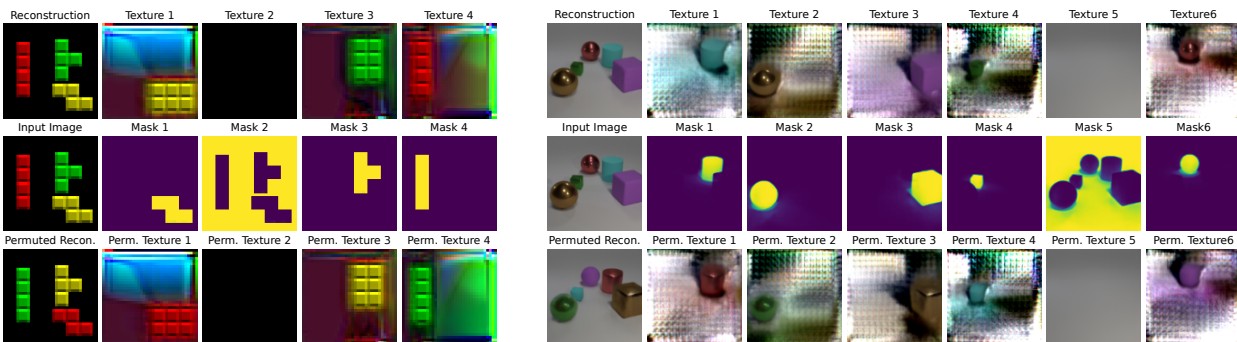

(a) Permutation applied to $S^{\text{text}}$: 4-2-1-3.

(b) Permutation applied to $S^{\text{text}}$: 6-4-2-1-5-3.

Figure 5: Compositional capabilities of DISA on (**a**) Tetrominoes and (**b**) CLEVR6. The central row contains the input image and the predicted object masks. At the top, the reconstructed image and object textures are shown (empty slots excluded), while the bottom row presents the reconstruction and object textures after interchanging texture vectors between objects. DISA shows strong compositional generalization, enabling reliable transferring of textures between objects while preserving shape, position, and scale information. Moreover, mask predictions are highly accurate with clear background separation.

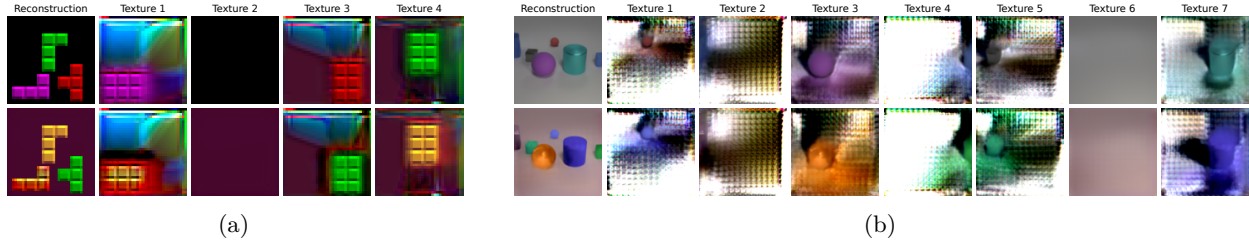

(a)

(b)

Figure 6: Texture generation capabilities of DISA on (**a**) Tetrominoes and (**b**) CLEVR6. The top row contains the reconstructed input image and object texture. At the bottom, the same visualization is presented after replacing the texture vectors with ones sampled from $\mathcal{N}(\mu_{\text{tex}}, 0.035)$, with $\mu_{\text{tex}}$ being the mean of the texture vectors encoded from the input image. DISA's learned latent space allows us to swap in completely new textures for specific objects independently of their shape, location, and scale, demonstrating the generative capabilities of the model.

objects while preserving shape, position, and scale information. Moreover, we see highly accurate mask predictions with clear background separation.

In Figure 6 we show the texture generation results of DISA on Tetrominoes and CLEVR6. On Tetrominoes (Figure 6a), the decoded textures maintain the original shapes but switch appearance. From the first slot, it is also visible that we can generate coherent out-of-distribution textures, as the dataset only contains objects with plain colors, while the one we obtain mixes red and yellow. Concerning CLEVR6 (Figure 6b), we find again that DISA is able to precisely adapt the sampled textures to the fixed shapes, positions, and scales of the objects in the image. Figure 6 shows therefore that the learned representations of DISA also allow for generative capabilities: we can swap in completely new textures for specific objects independently of their shape, location, and scale.

We show in Figure 7 the shape generation results of DISA on Tetrominoes and CLEVR6. Note that this task is quite difficult: first, the number of unique shapes is strongly limited in these synthetic datasets; second, without changing the scale of the reference frame, the generated shape is restricted to one that fits the original frame. On Tetrominoes (Figure 7a), the sampled shapes are roughly defined but close to those present in the training data, with exceptions such as that of the cyan object on the bottom left. On CLEVR6 (Figure 7b)

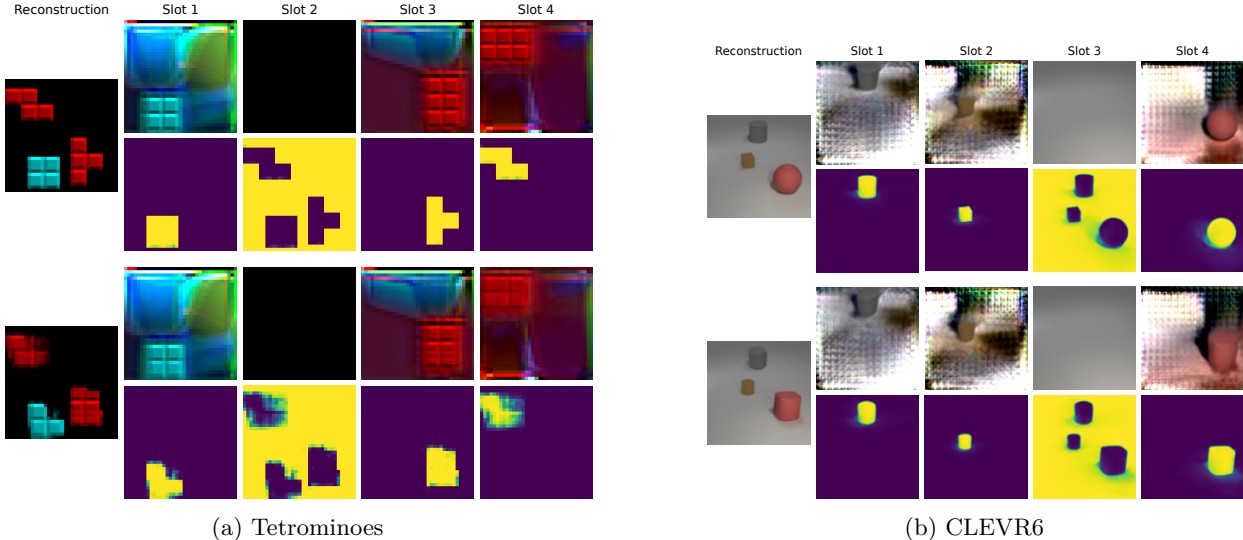

(a) Tetrominoes          (b) CLEVR6

Figure 7: Shape generation capabilities of DISA on (**a**) Tetrominoes and (**b**) CLEVR6. In each of the 2 images, the first pair of rows shows final reconstruction (middle left), object textures (first row), and shapes (second row) obtained after encoding and decoding an input image. The second pair of rows presents final reconstructions (middle left) and individual object textures (first row) and shapes (second row) after sampling new shape vectors. On Tetrominoes, the sampled shapes are roughly defined but close to those present in the training data with exceptions such as that of the cyan object on the bottom left, while on CLEVR6 the generated shapes are almost identical to those present in the training data. The difficulty of this task is mainly due to the strongly limited number of unique shapes in these synthetic datasets and to the fact that, without changing the scale of the reference frame, the generated shape is restricted to one that fits the original frame.

the generated shapes are almost identical to those present in the training data. A higher standard deviation during the sampling phase can lead to a larger variety in the generated shapes, while at the same time can strongly degrade their quality.

## 6 Conclusion

In this work, we introduce a novel approach named Disentangled Slot Attention (DISA). DISA explicitly disentangles the groups of features responsible for encoding texture, shape, position, and scale in object-centric representations. Our method results in a latent space characterized by four non-overlapping subsets of its dimensions, known prior to the start of the training process. To train DISA, we employ the commonly used reconstruction loss along with an additional simple regularization of the latent space, aiming to support the architecture in preventing texture and shape information from flowing within unrelated components. Our quantitative experiments demonstrate that, in most cases, DISA achieves the desired disentanglement of texture and shape components while being competitive with or outperforming the baselines in image decomposition and reconstruction quality. Additionally, we show compositional and generative capabilities derived from the disentanglement through qualitative experiments.

There are several directions for future work. First, as seen on CLEVR6 and CLEVRTex, more complex textures can cause some leakage of texture information into shape-related components. A possible direction to address this problem is to exploit a stronger filter (than the Sobel filter) to further remove texture information from the objects. Vice versa, one may also look into additional strategies to prevent shape information from flowing into texture dimensions. Finally, DISA does not tackle the further disentanglement of the features inside the explicitly separated groups, which could also be a promising direction for future work.

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

## A  Sobel Filter

The Sobel filter (Sobel & Feldman, 1968) uses two $3 \times 3$ filters to approximate the horizontal and vertical derivatives of an image:

$$\mathcal{K}_x = \begin{bmatrix} +1 & 0 & -1 \\ +2 & 0 & -2 \\ +1 & 0 & -1 \end{bmatrix} \quad \text{and} \quad \mathcal{K}_y = \begin{bmatrix} +1 & +2 & +1 \\ 0 & 0 & 0 \\ -1 & -2 & -1 \end{bmatrix} . \tag{8}$$

A 2-dimensional convolution operation with $\mathcal{K}_x$ and $\mathcal{K}_y$ is applied on the input image $\mathcal{X}$, producing two maps of the approximated horizontal and vertical derivatives at each pixel, respectively $\mathcal{G}_x = \mathcal{K}_x * \mathcal{X}$ and $\mathcal{G}_y = \mathcal{K}_y * \mathcal{X}$. Figure 8b shows an example of the resulting $\mathcal{G}_x$ and $\mathcal{G}_y$ of the grayscale image 8a. To obtain the final filtered input containing the detected edges, the maps are aggregated as

$$\mathcal{G} = \sqrt{\mathcal{G}_x^2 + \mathcal{G}_y^2} . \tag{9}$$

The filtered image associated with the previous example is shown in figure 8c.

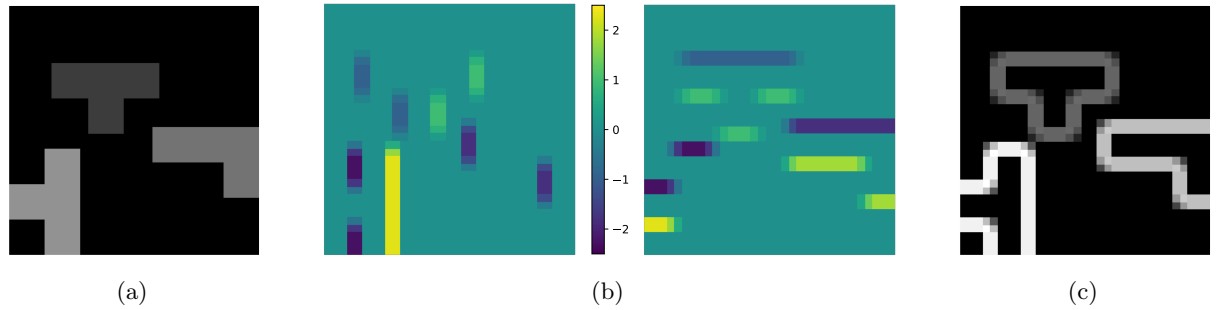

|  (a)  |  (b)  |  (c)  |

Figure 8: Example of the Sobel filter applied to a simple grayscale image. (**a**) Input image $\mathcal{X}$. (**b**) Maps of the approximated horizontal ($\mathcal{G}_x$, left) and vertical ($\mathcal{G}_y$, right) derivatives of $\mathcal{X}$ at each pixel. (**c**) Filtered input image $\mathcal{G}$ computed combining $\mathcal{G}_x$ and $\mathcal{G}_y$ following Equation 9.

When dealing with RGB images, there are two possible approaches. The first is to simply convert the image to grayscale before applying the Sobel filter. The second, which we employ in our method, consists in applying the filter independently to each channel, then averaging the three resulting filtered images.

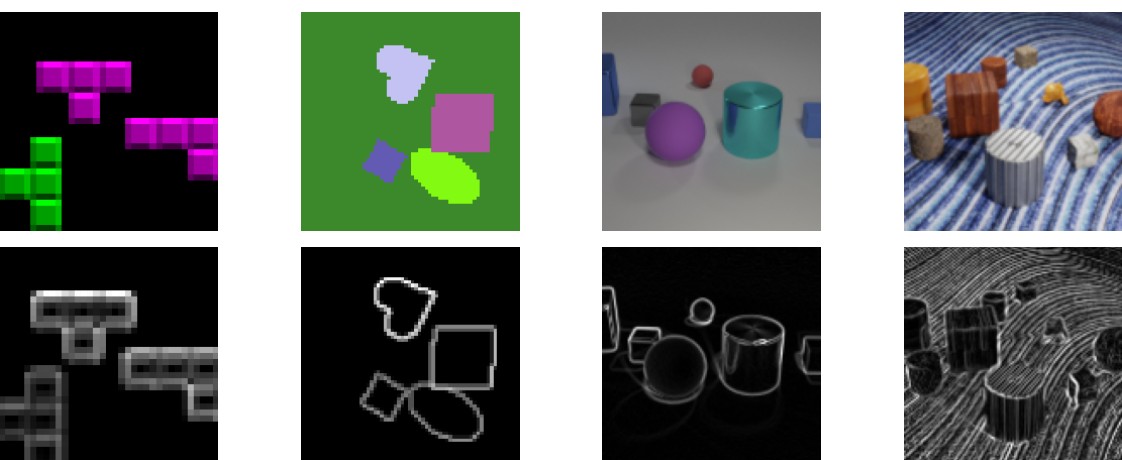

Figure 9: Example of the Sobel filter applied to an image from each of the considered datasets. The top row shows the input images, while the bottom row displays the outputs of the Sobel filter.

# B    Datasets

**Tetrominoes**    Tetrominoes consists of $35 \times 35$ RGB images presenting 3 "Tetris"-like blocks and a black background. The blocks are characterized by one of 6 possible colors (red, green, blue, yellow, magenta, cyan) and a shape and orientation within 19 possible. Figure 10a shows a few examples extracted from the dataset. The tetrominoes can appear anywhere in the image, but cannot overlap with each other. Furthermore, multiple shapes/orientations/colors of the same kind can be included in a single sample. We employ a total of 60K samples for the training set and 320 for the test set, in line with Greff et al. (2019); Locatello et al. (2020).

**Multi-dSprites**    Multi-dSprites is a dataset based on dSprites (Matthey et al., 2017), where three types of shapes, namely oval, heart, and square, are represented over a plain background. Both the sprites and the background are colored with randomly sampled RGB values. The number of entities in the image can vary from 1 to 4 excluding the background, and partial occlusion can be present. Each image has a resolution of $64 \times 64$. Examples from the Multi-dSprites dataset are visualized in Figure 10b. Again, we employ a total of 60K samples for the training set and 320 for the test set, as in Greff et al. (2019); Locatello et al. (2020).

**CLEVR**    CLEVR, introduced by Johnson et al. (2017), is a collection of $240 \times 320$ images of 3D scenes containing a number of objects ranging from 3 to 10 (excluding the background). There are three types of shapes, i.e. cube, sphere, and cylinder, two possible sizes (small and large), eight colors (gray, red, blue, green, brown, purple, cyan, and yellow), and two materials (shiny "metal" and matte "rubber"). Some samples from CLEVR are presented in Figure 10c. We also employ a filtered version of this dataset, called CLEVR6, that has a maximum of 6 entities in the scenes, background excluded. The number of training samples, in this case, is 70K, while the test samples are 15K. In all our experiments, a center crop of $192 \times 192$ is applied, followed by a resize to a resolution of $128 \times 128$.

**CLEVRTex**    CLEVRTex (Karazija et al., 2021) consists of 50K $240 \times 320$ images (cropped and resized to $128 \times 128$ as in CLEVR), 40K of which are training samples while the remaining 10K are equally split into validation and testing sets. As in CLEVR, each image contains between 3 and 10 objects plus the background. For each object (background included), the material is sampled from a collection of 60 materials (more complex than those in CLEVR). Objects can appear under 4 different shapes (cube, cylinder, sphere, and non-symmetric monkey head) and 3 distinct scales (small, medium, and large). In addition, the dataset presents several variations, such as CAMO and OOD. CAMO is a collection of 20K test images where all the objects, including the background, share the same material. OOD consists of 10K test samples with 25 materials and 4 shapes (cone, torus, icosahedron, and a teapot) not contained in the training split. In Figure 10d, we show some examples extracted from the dataset.

# C    Additional Discussion on Related Work

This section further describes two approaches (mentioned in Section 3) dealing with explicit disentanglement of shape and texture, highlighting the differences with our work.

**Disentangling 3D Prototypical Networks For Few-Shot Concept Learning**    Prabhudesai et al. (2020) proposes an interesting approach (D3DP) to explicitly disentangle shape and texture information in an object-centric way. Despite being developed to address the same challenge as DISA, it presents multiple differences from our work. Similar to our architecture, D3DP uses two different encoders to individually embed shape and texture information. However, it employs a single decoder -based on adaptive instance normalization (Huang & Belongie, 2017)- to simultaneously reconstruct texture and shape. In our work, using two separate decoders is considered very important: by decoding the masks based solely on the shape-related part of the slots, we ensure that shape information is present in those features. Another substantial difference between the two approaches is how they separate objects into distinct vectors or slots. DISA, along with other Slot Attention-based approaches, carries out this task entirely in an unsupervised manner. D3DP, on the other hand, relies on ground-truth 3D bounding boxes, essential to decompose the scene into objects. Additionally, the training strategy of D3DP assumes the availability of multiple viewpoints for a given static

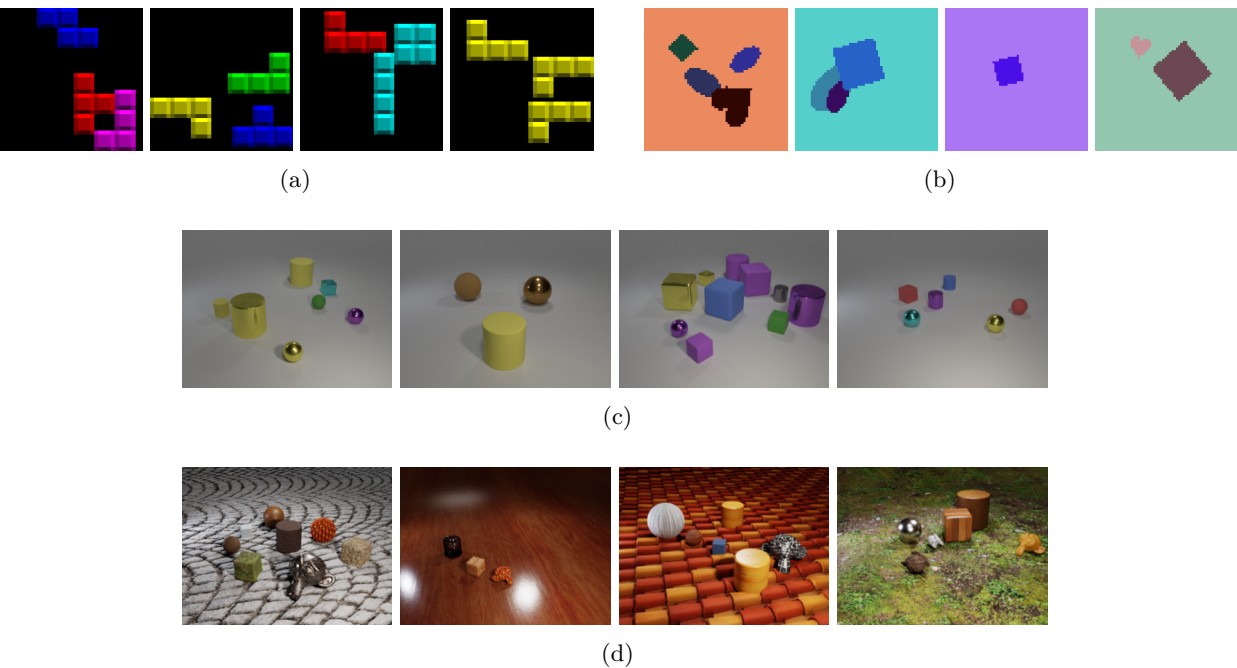

Figure 10: Samples extracted from both train and test partitions of (**a**) Tetrominoes, (**b**) Multi-dSprites, (**c**) CLEVR, and (**d**) CLEVRTex.

scene. This is because the encoder should learn to predict a 2D point of view of a scene from a distinct 2D viewpoint, passing through a complete 3D feature map of the scene. Such a strong assumption makes D3DP hard to apply to real-world dynamic scenarios, in which it is unlikely that an agent can observe the exact same scene from more than one viewpoint. On the contrary, DISA (and Slot Attention-based models in general) does not depend on this assumption. Finally, the authors of D3DP do not quantitatively evaluate the disentanglement of texture and shape, for example, through regular and inverse property prediction tasks (as in our work).

**Unsupervised Part-Based Disentangling of Object Shape and Appearance**  Lorenz et al. (2019) introduces an unsupervised method to disentangle the shape and texture components in object representations. The first difference with DISA is that they work on single object images belonging to a unique class (e.g., people, dogs, or birds). In fact, they extract a fixed number of object parts (e.g., left arm or head), each encoded by a pair of texture and shape vectors. In contrast, DISA extracts disentangled vectors representing individual objects in a multi-object scene, where the number of objects is variable. Although both considered architectures rely on two separate encoders to represent shape and texture, there are key differences to be noted. Their method exploits spatial transformations to help prevent shape information from being included in the texture components. Instead, DISA extracts texture information from the original input image and relies on the presence of the necessary shape information in the shape-related features (enabled by the mask decoder, as mentioned in the previous paragraph). The assumption is that, as we already provide the texture decoder with the shape vector, including shape information in the texture vector would be redundant and thus unnecessary (the variance regularization also helps, but Figure 14 suggests that the architecture plays a more important role). In future work, it could be interesting to study whether augmenting our approach with these spatial transformations would lead to even better results. Another difference is in the appearance transformation. In our work, we remove part of the texture information (mainly color) with a Sobel filter to bias the shape encoder toward encoding only shape information. Their strategy is instead to change the brightness, contrast, and hue of the image fed to the shape encoder. Both ways are limited in that they do not remove or change the structure of a texture, but only its color. A further distinction is that the shape encoder of DISA represents the shape information of an object as a single low-dimensional vector, while that employed by Lorenz et al. (2019) encodes a specific subpart of the object as an activation map over the input

image (similar to those in the Slot Attention mechanism). Moreover, their approach adopts a single decoder to reconstruct the entire input image, while DISA uses two decoders to individually predict the mask and texture of every object. Finally, as D3DP, this work does not present any quantitative analysis on shape and texture disentanglement but only a qualitative study on compositionality.

# D Training Setup

## D.1 Object Discovery

SA, ISA, and DISA are trained as autoencoders on Tetrominoes, Multi-dSprites, CLEVR6, and CLEVRTex for an equal number of steps. A batch size of 64 and a learning rate of $4 \times 10^{-4}$ (using Adam as optimizer (Kingma & Ba, 2014)) are employed on all datasets for all the architectures. An initial (linear) warm-up for 10K steps and an exponential decay schedule (from start to end) are applied to the learning rate. On Tetrominoes, we perform roughly 100K steps (107 epochs $\times$ 938 batches), on Multi-dSprites around 250K (267 epochs $\times$ 938 batches), on CLEVR6 nearly 150K (274 epochs $\times$ 547 batches), and on CLEVRTex exactly 150K (240 epochs $\times$ 625 batches). The mean squared error is used as the reconstruction loss function.

The number of iterations in the Slot Attention mechanism is fixed to 3, while the number of slots is set to 4 on Tetrominoes, 6 on Multi-dSprites, 7 on CLEVR6, and 11 on CLEVRTex. Moreover, for SA the slots are initially sampled from a learned distribution, while for ISA and DISA are initialized as learnable vectors. When we set the initial slots as learnable vectors, hence on ISA and DISA models, we decide to employ the bi-level optimization from Jia et al. (2022). We set the dimensionality of the slot vectors to 64 (excluded position and scale factors) in all cases and, in DISA, we define the texture components as the first half and the mask components as the second half (32 each) of a representation. With DISA, we always set the initial position and scale vectors as learnable embeddings, while with ISA only on Multi-dSprites and CLEVRTex. On Tetrominoes and CLEVR6 we sample those as done in the original paper for the two specific datasets. Furthermore, again in line with the original paper, ISA does not use scale vectors on Tetrominoes.

With ISA and DISA, we clip the norm of gradients to 0.05 (as in ISA's original implementation). Finally, on Tetrominoes and Multi-dSprites, we apply the variance regularization only on $S^{\text{tex}}$ with $\lambda = 0.32$ since the beginning of the training, while on CLEVR6 and CLEVRTex we apply it after the warm-up on both $S^{\text{tex}}$ and $S^{\text{shape}}$ (as in Equation 7) with $\lambda = 0.05$.

On Tetrominoes and CLEVR, DISA tends to learn masks that, instead of precisely segmenting an object, select areas of arbitrary shape around the entities. Despite maintaining good decomposition capabilities as the objects get correctly divided into separate slots, this behavior can hinder the shape information from being encoded in the shape components. Consequently, the texture components would have to include shape information, leading to an incorrect disentanglement. We can notice that the issue seems to arise when a dataset is characterized by a fixed background, but not with a dynamic one as in Multi-dSprites and CLEVRTex. The simple introduction of random plain noise added to the input images (Figure 11) can avoid this problem: in fact, the intuition behind the solution is to induce slight changes in the background texture, so that it is no longer fixed, while however not heavily altering the foreground objects. We hence adopt this augmentation strategy when training our DISA models on Tetrominoes and CLEVR6. Additional experiments are required to gain more insights on this, for instance, to understand whether fixed non-plain backgrounds are as affected as fixed plain ones.

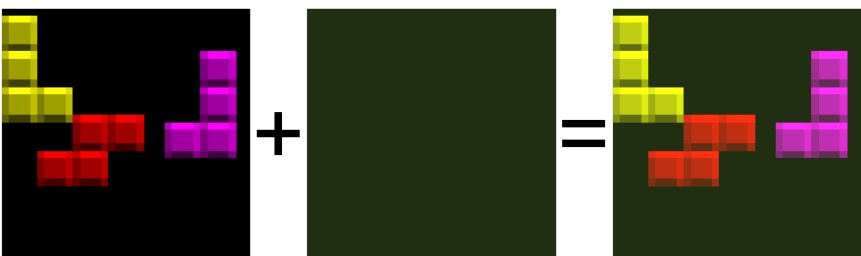

Figure 11: Example of the plain noise augmentation applied on an image from Tetrominoes. The augmentation induces slight changes in the color of a fixed background, turning it dynamic and helping the model avoid suboptimal solutions.

### D.2 Property Prediction

Each MLP used in our property prediction experiments has a single hidden layer of 256 nodes activated with Leaky ReLU. The input layer has 32 nodes because only texture or shape components are used to predict the property of an object. On every dataset, we perform about 10k steps with a batch size of 64 and a learning rate set to $1 \times 10^{-3}$ (the optimizer is again Adam). With categorical variables we employ the cross-entropy loss, while with numerical the MSE. Moreover, on all datasets, the last 1000 samples are excluded from the training set and utilized as validation set.

## E    Disentanglement Analysis

### E.1    Training and Validation Curves

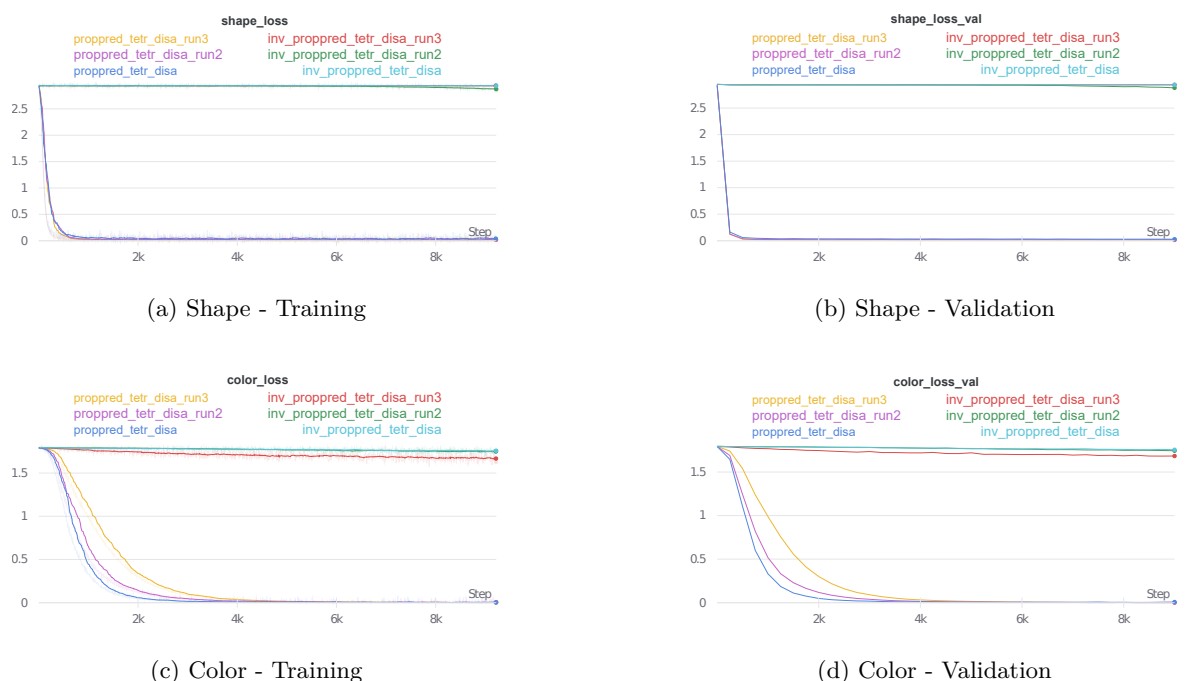

(a) Shape - Training

(b) Shape - Validation

(c) Color - Training

(d) Color - Validation

Figure 12: Training and validation curves on Tetrominoes. Each plot represents the cross-entropy loss (y-axis) as a function of the training steps (x-axis).

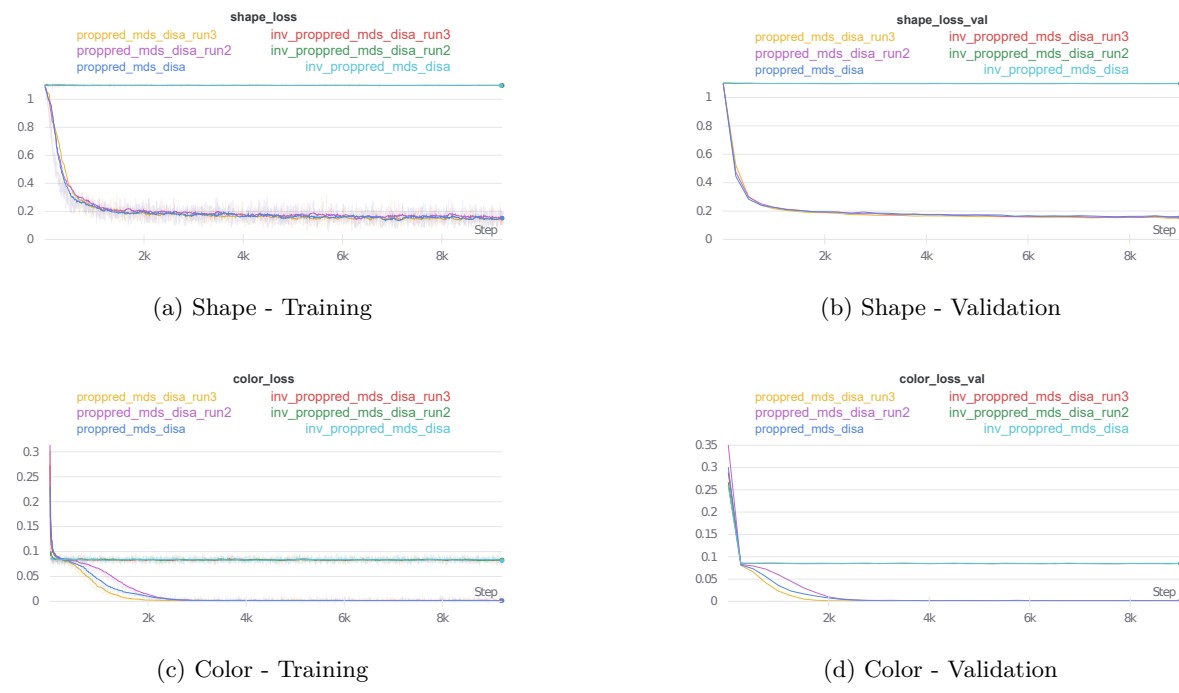

Figure 13: Training and validation curves on Multi-dSprites. The first two plots (**a** and **b**) represent the cross-entropy loss (y-axis) as a function of the training steps (x-axis). The last two plots (**c** and **d**) represent the mean squared error (y-axis) as a function of the training steps (x-axis).

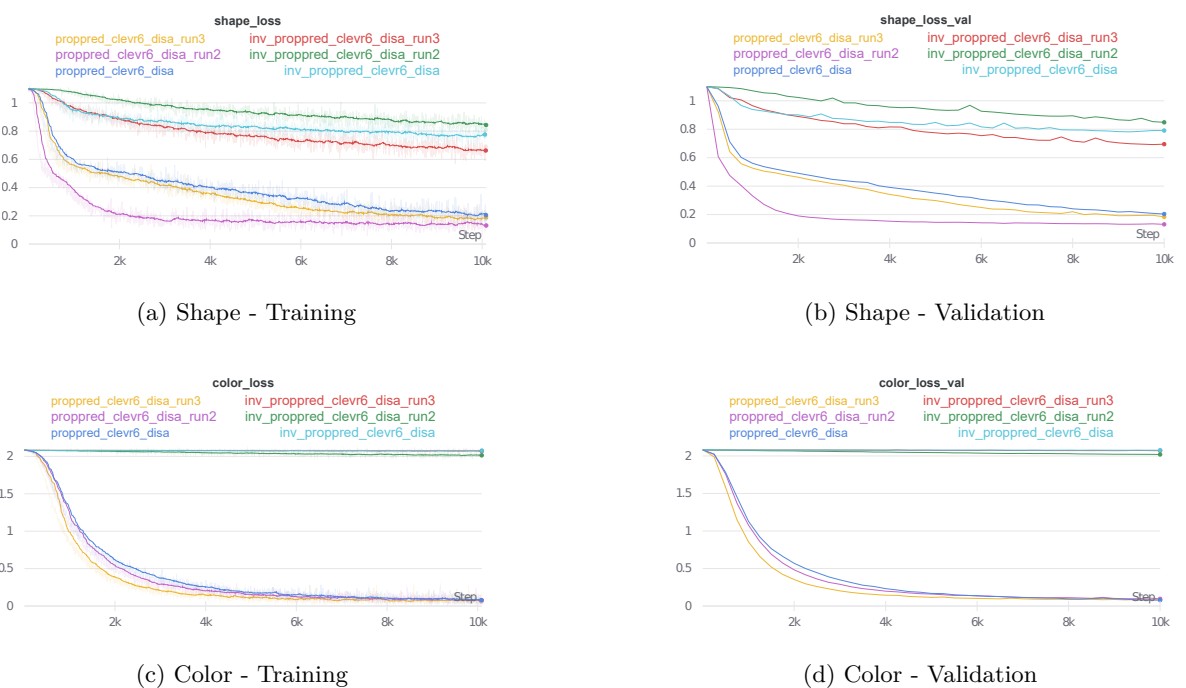

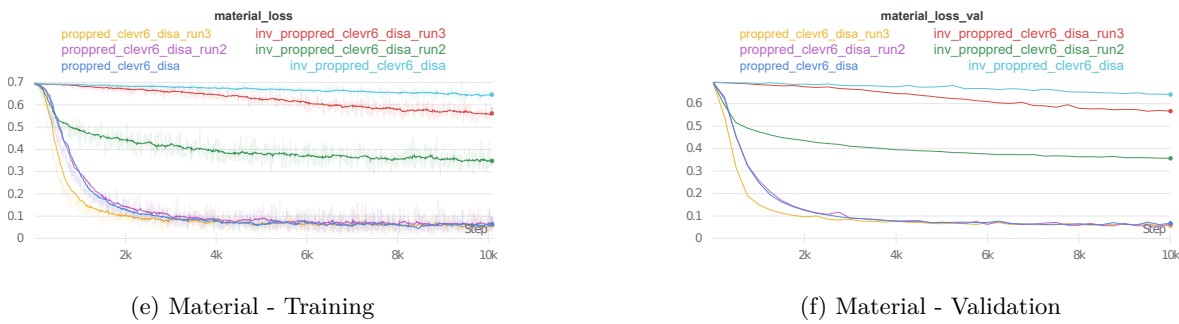

(e) Material - Training                    (f) Material - Validation

Figure 14: Training and validation curves on CLEVR6. Each plot represents the cross-entropy loss (y-axis) as a function of the training steps (x-axis).

# F    Additional Experimental Results

## F.1    Ablation: Impact of Variance Regularization and Sobel Filter on Texture and Shape Disentanglement

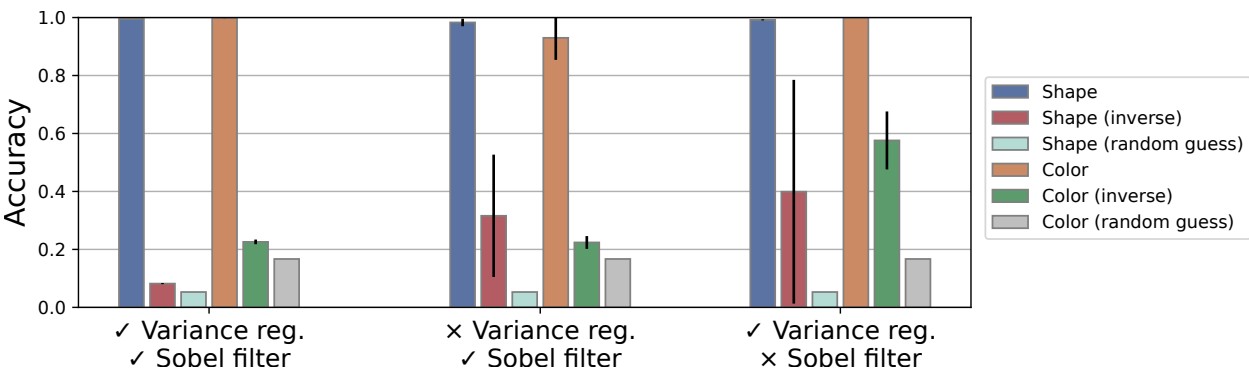

Figure 15: Quantitative results of DISA on the regular and inverse property prediction tasks. We report the measurements of DISA with Sobel filter and variance regularization (left, shown in Figure 3), without the use of the variance regularization (center), and without Sobel filter (right) on Tetrominoes. The predicted properties are shape and color, both categorical variables. The prediction accuracy is employed and compared with a baseline random guess. We report mean and stddev over 3 seeds.

### F.2 Compositional Results

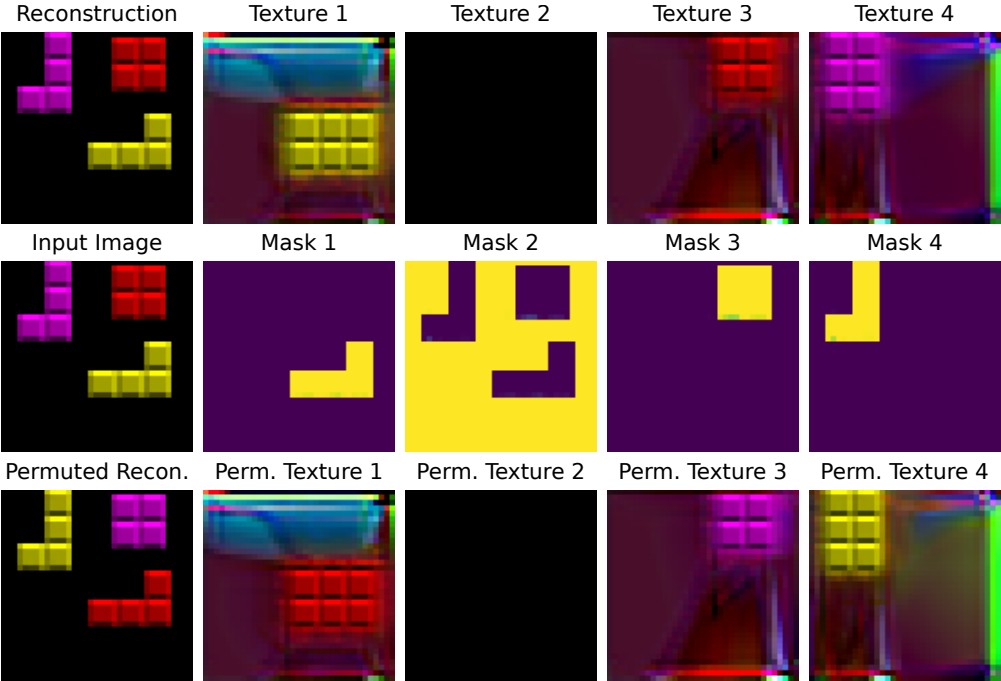

Figure 16: Compositionality on Tetrominoes. Permutation applied to $S^{\text{text}}$: 3-2-4-1.

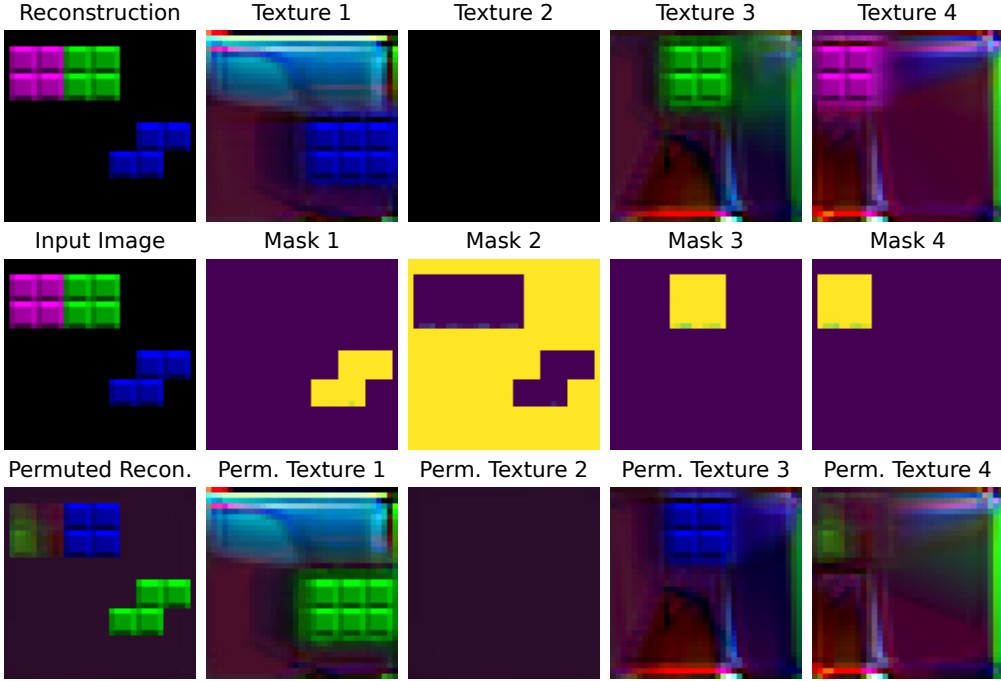

Figure 17: Compositionality on Tetrominoes. Permutation applied to $S^{\text{text}}$: 3-4-1-2.

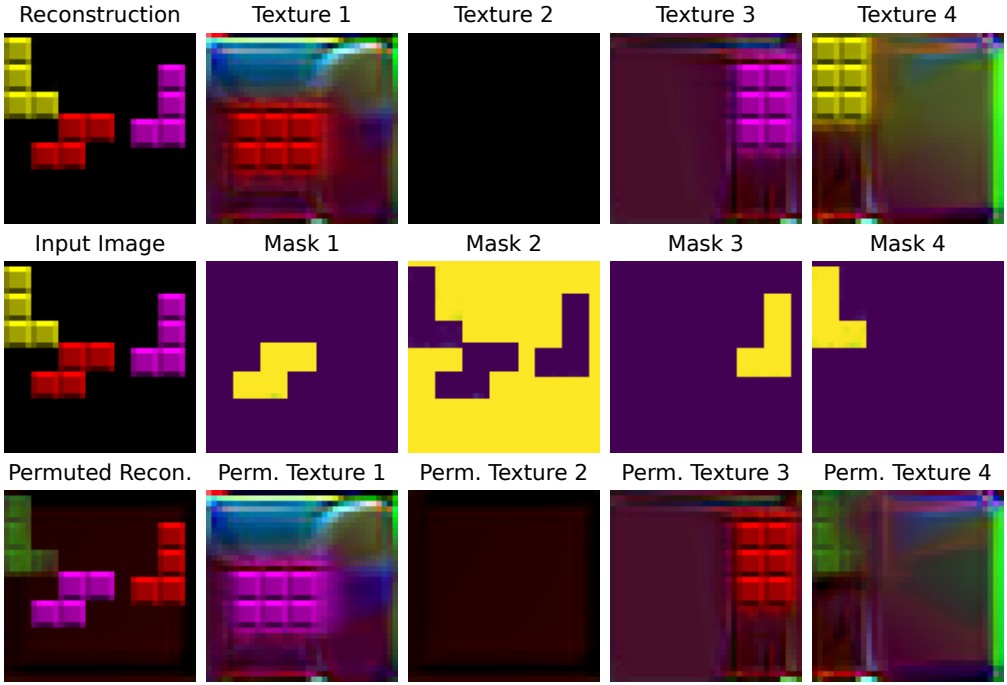

Figure 18: Compositionality on Tetrominoes. Permutation applied to $S^{\text{text}}$: 3-4-1-2.

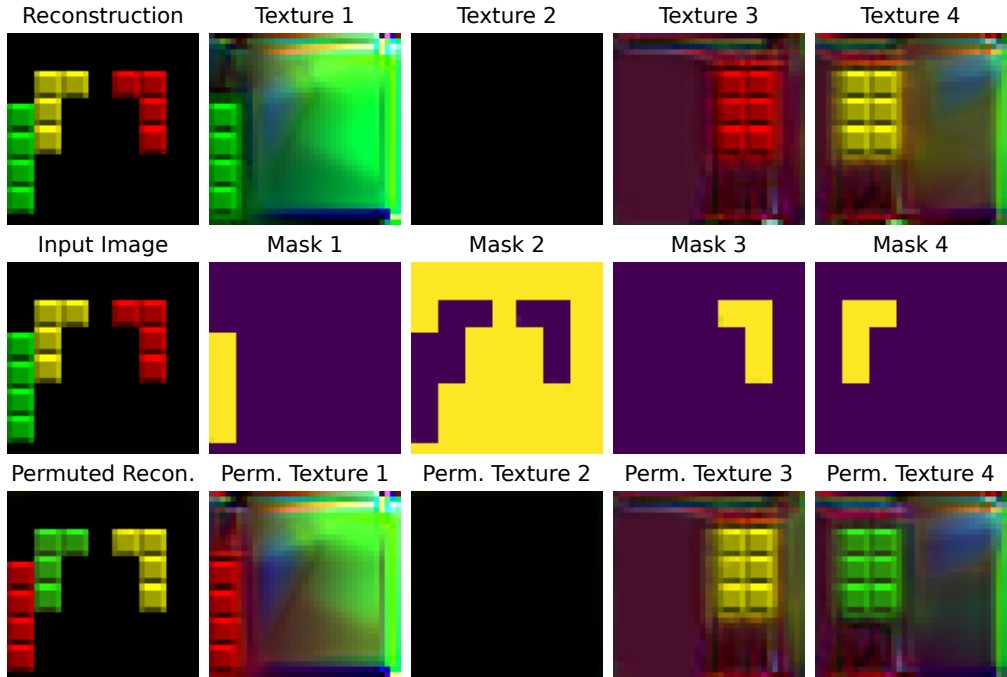

Figure 19: Compositionality on Tetrominoes. Permutation applied to $S^{\text{text}}$: 3-2-4-1.

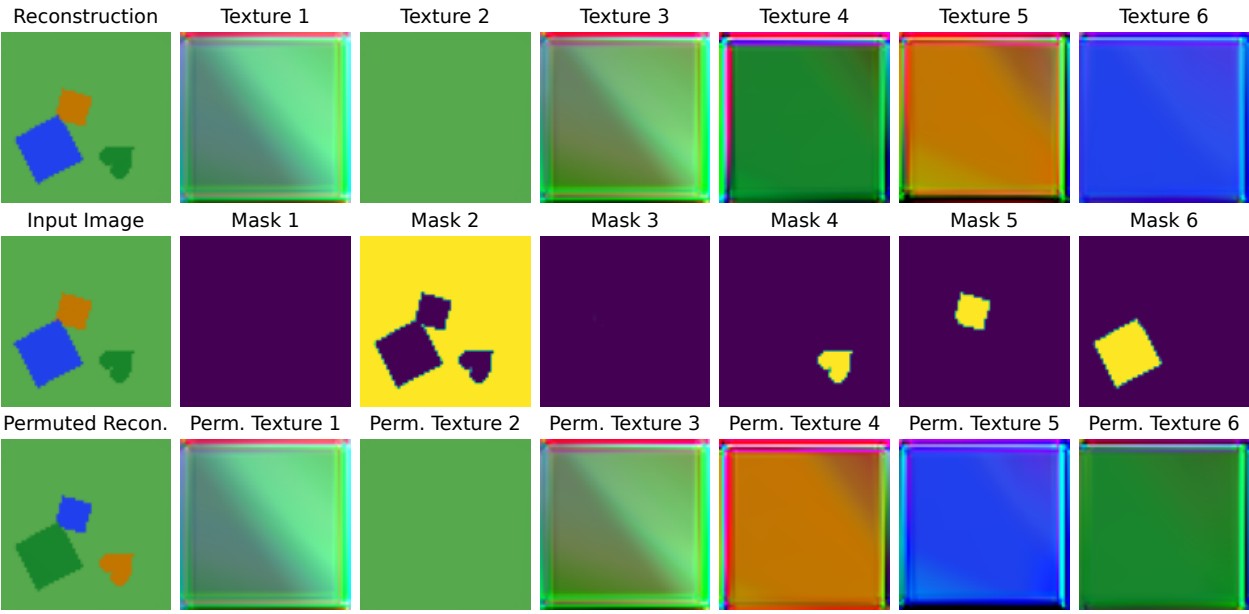

Figure 20: Compositionality on Multi-dSprites. Permutation applied to $S^{\text{text}}$: 1-2-3-5-6-4.

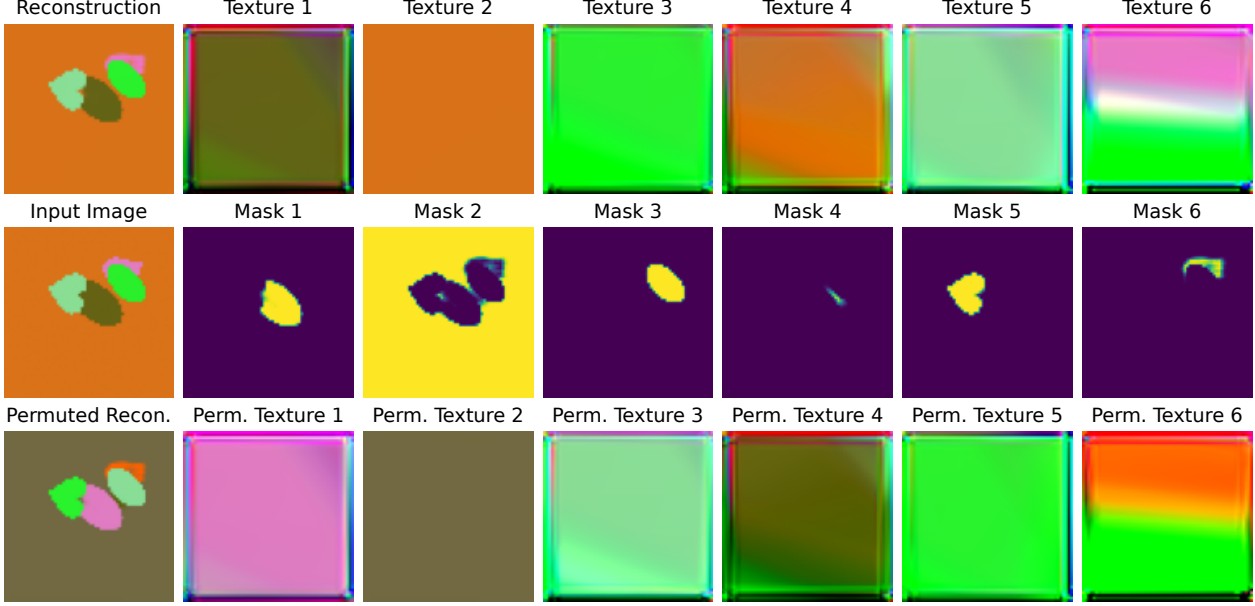

Figure 21: Compositionality on Multi-dSprites. Permutation applied to $S^{\text{text}}$: 6-1-5-1-3-4.

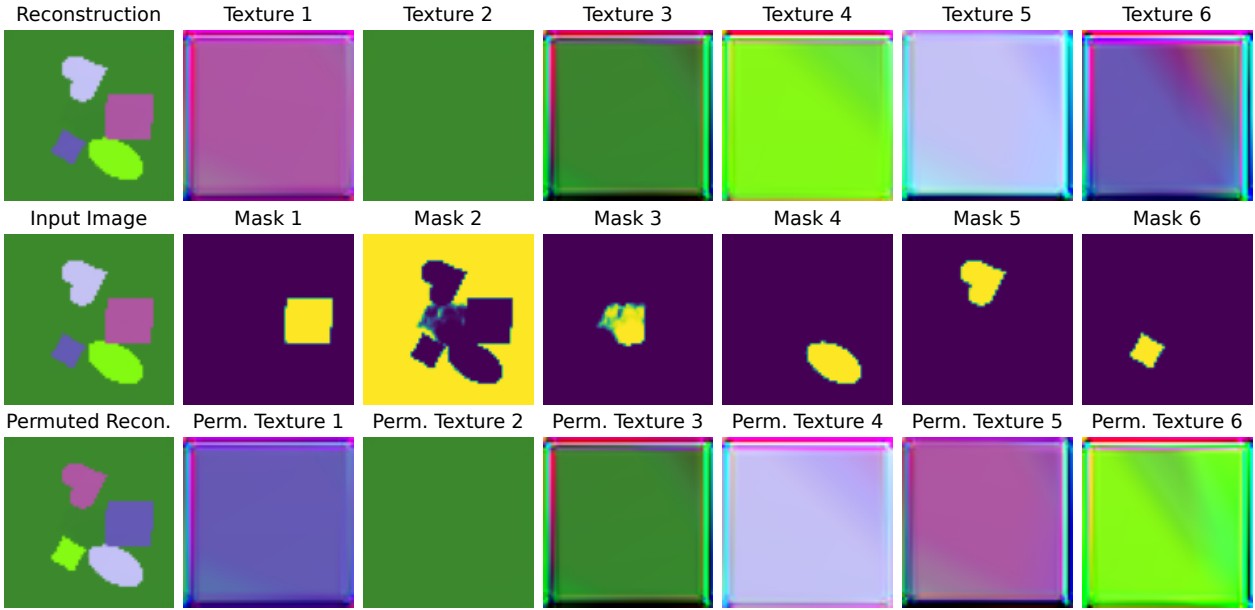

Figure 22: Compositionality on Multi-dSprites. Permutation applied to $S^{\text{text}}$: 6-2-3-5-1-4.

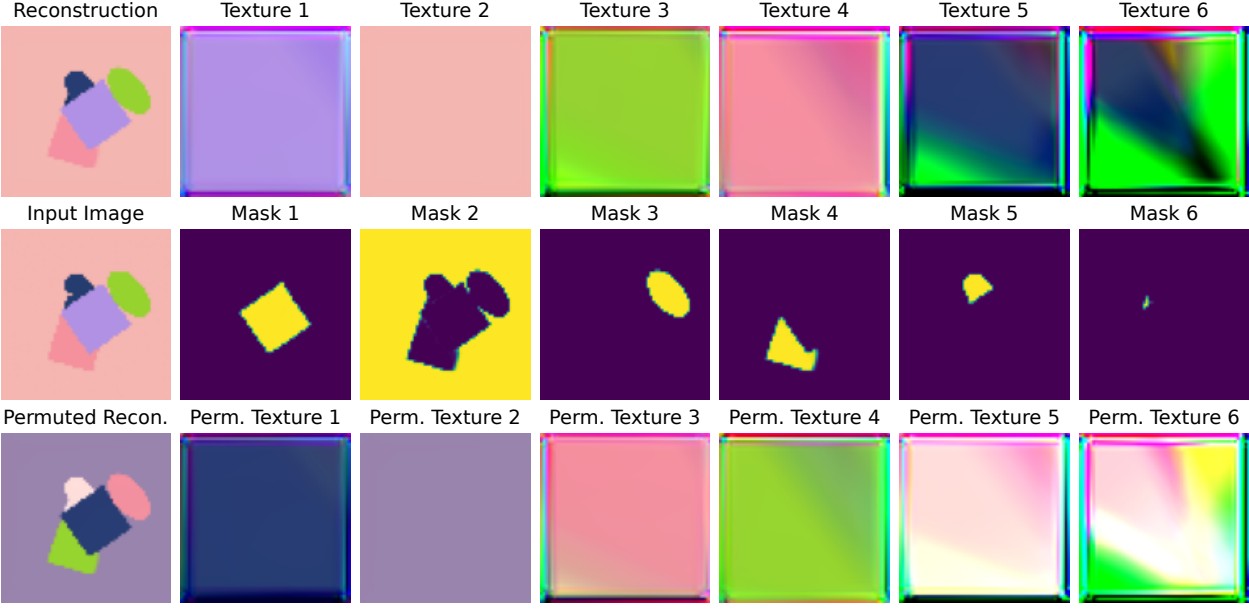

Figure 23: Compositionality on Multi-dSprites. Permutation applied to $S^{\text{text}}$: 5-1-4-3-2-2.

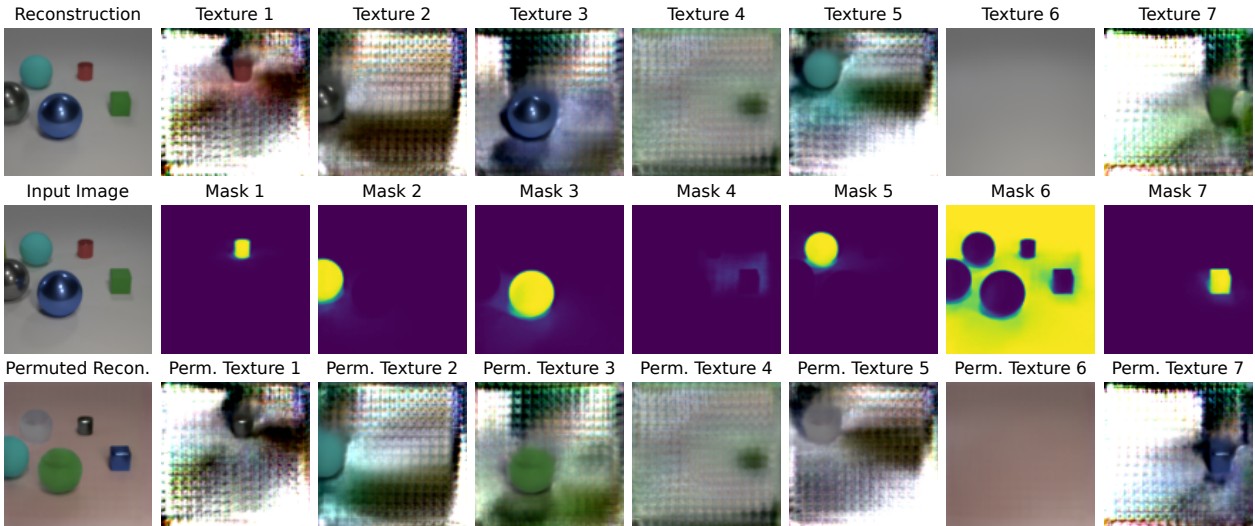

Figure 24: Compositionality on CLEVR6. Permutation applied to $S^{\text{text}}$: 2-5-7-4-6-1-3.

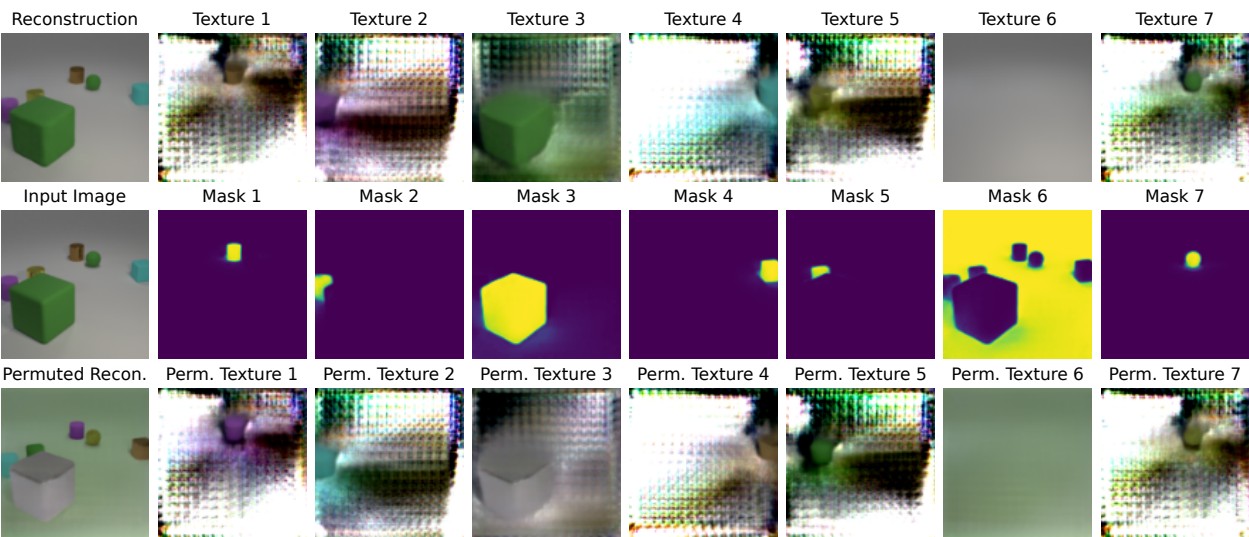

Figure 25: Compositionality on CLEVR6. Permutation applied to $S^{\text{text}}$: 2-4-6-1-3-7-5.

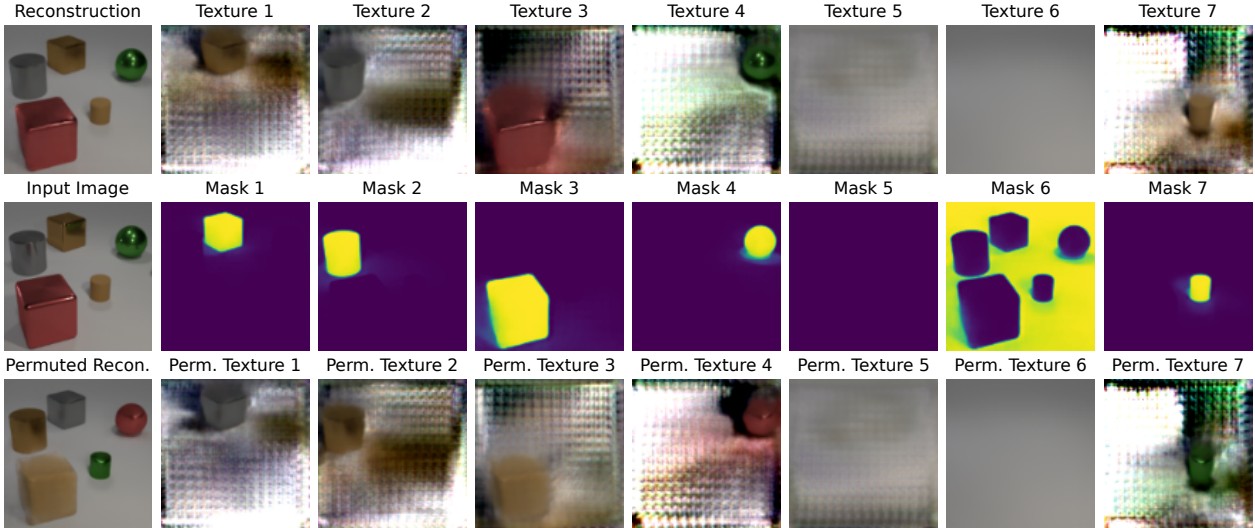

Figure 26: Compositionality on CLEVR6. Permutation applied to $S^{\text{text}}$: 2-1-7-3-5-6-4.

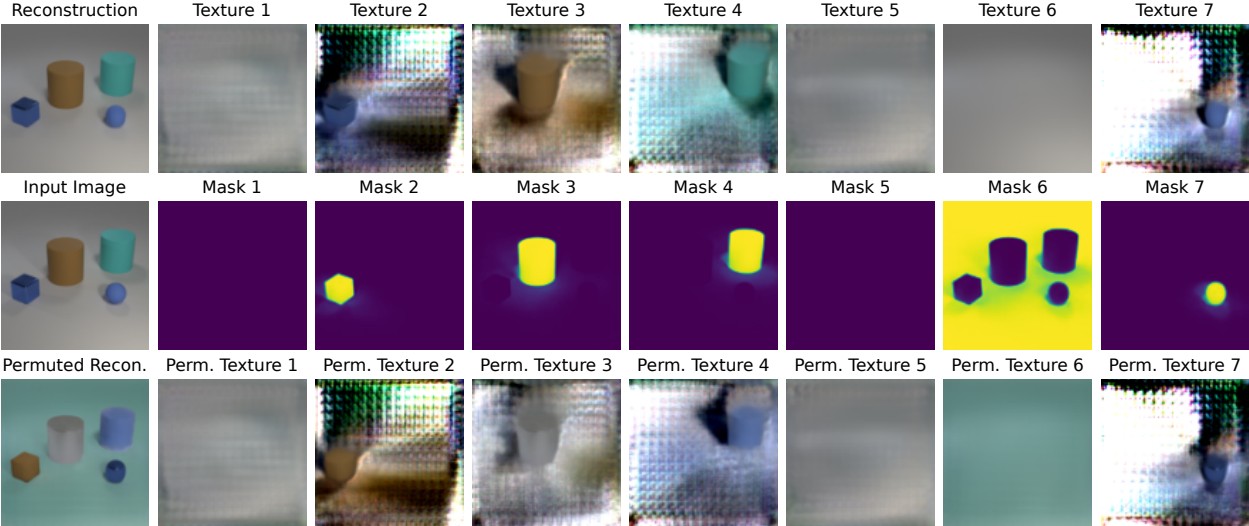

Figure 27: Compositionality on CLEVR6. Permutation applied to $S^{\text{text}}$: 1-3-6-7-5-4-2.

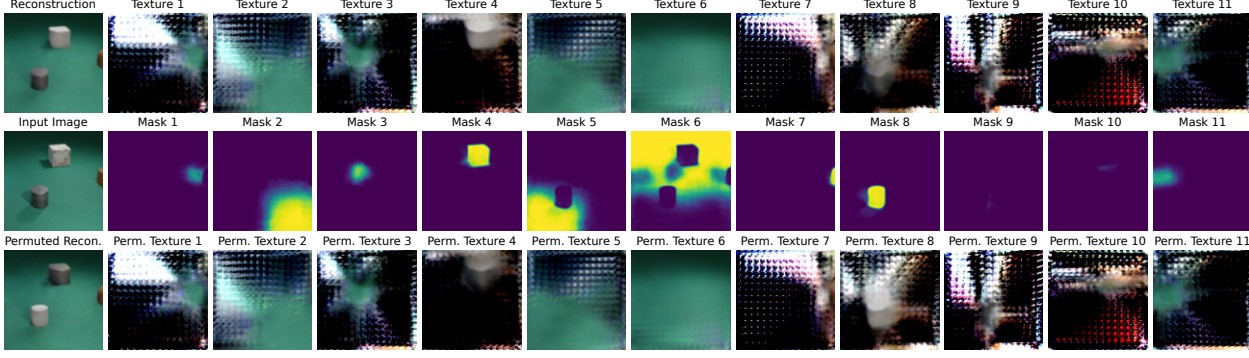

Figure 28: Compositionality on CLEVRTex. Permutation applied to $S^{\text{text}}$: 1-2-3-8-5-6-7-4-9-10-11.

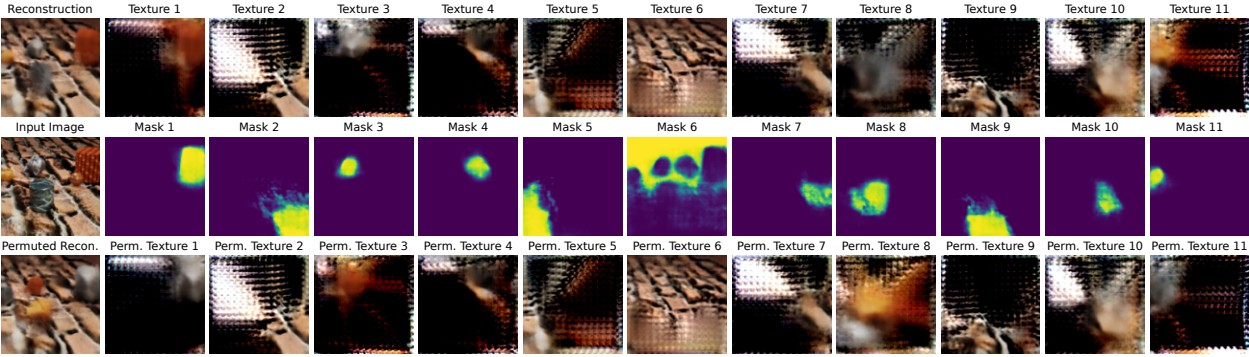

Figure 29: Compositionality on CLEVRTex. Permutation applied to $S^{\text{text}}$: 3-2-1-4-5-6-7-11-9-10-8.

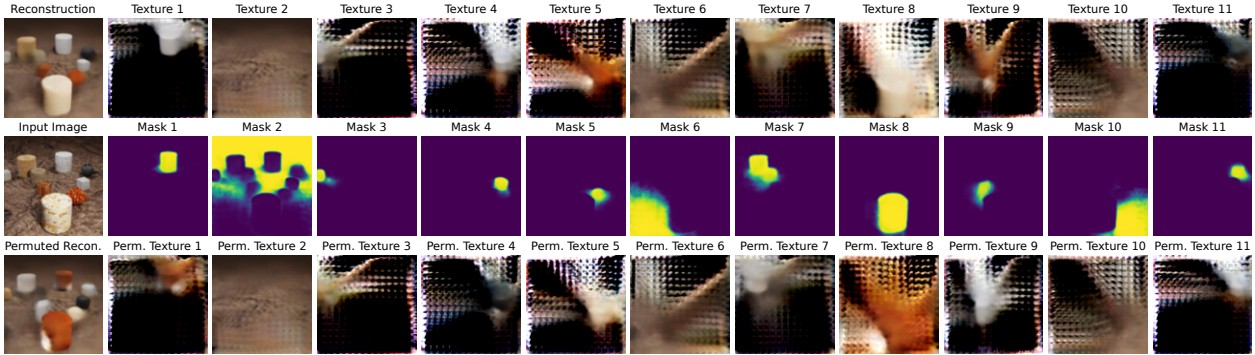

Figure 30: Compositionality on CLEVRTex. Permutation applied to $S^{\text{text}}$: 9-2-7-11-8-6-3-5-1-10-4.

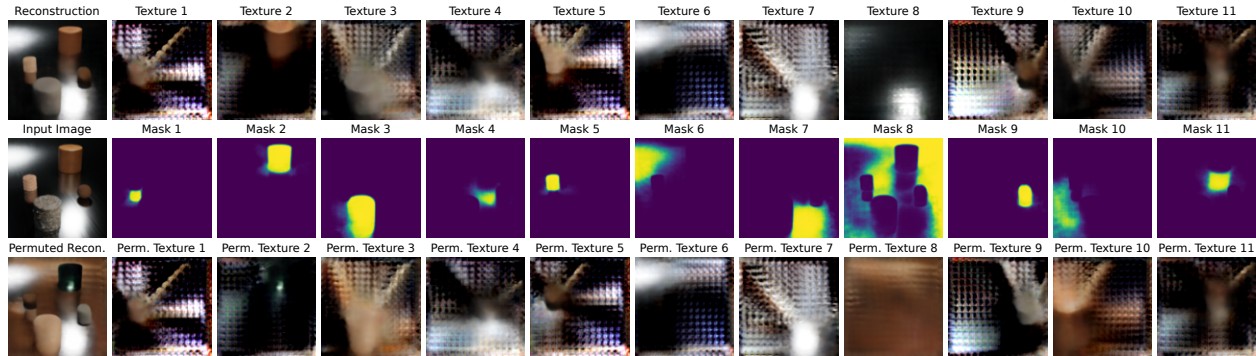

Figure 31: Compositionality on CLEVRTex. Permutation applied to $S^{\text{text}}$: 1-8-5-4-9-6-7-2-3-2-11.

### F.3 Generative Results

We include here results for texture and shape generation.

As anticipated, we sample the new textures from a Gaussian distribution centered on the mean of the texture vectors of the objects in a scene, with a standard deviation of 0.035 (manually tuned). Similarly, we sample new shape vectors from a Gaussian distribution centered on the mean of the shape vectors of the foreground objects in a scene, with a standard deviation manually tuned. On Tetrominoes and Multi-dSprites, the standard deviation is set to 0.1, while on CLEVR and CLEVRTex 0.01.

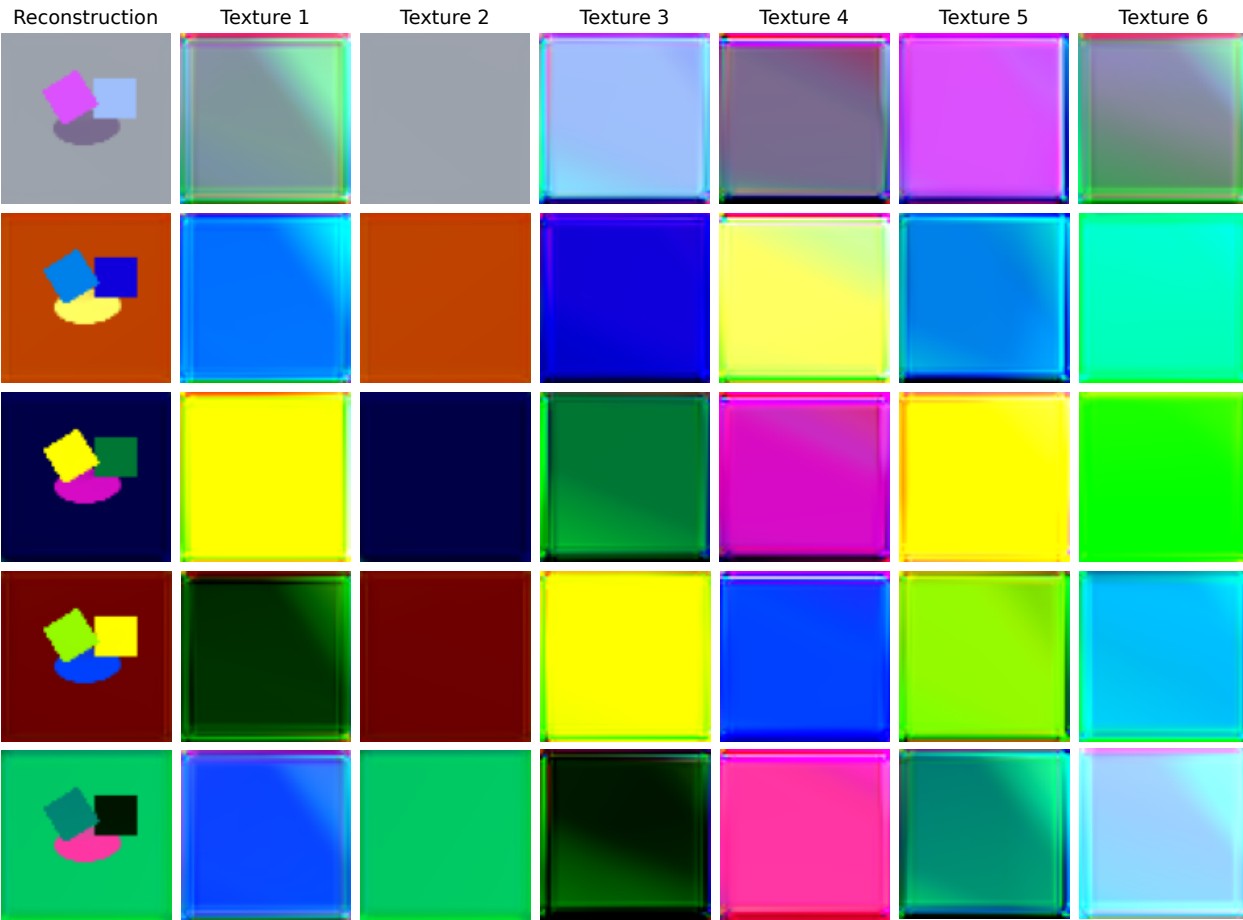

Figure 32: Texture generation on Multi-dSprites. The first row shows final reconstruction and object textures obtained after encoding and decoding an input image. The second to fifth rows present final reconstructions and individual object textures after sampling new texture vectors.

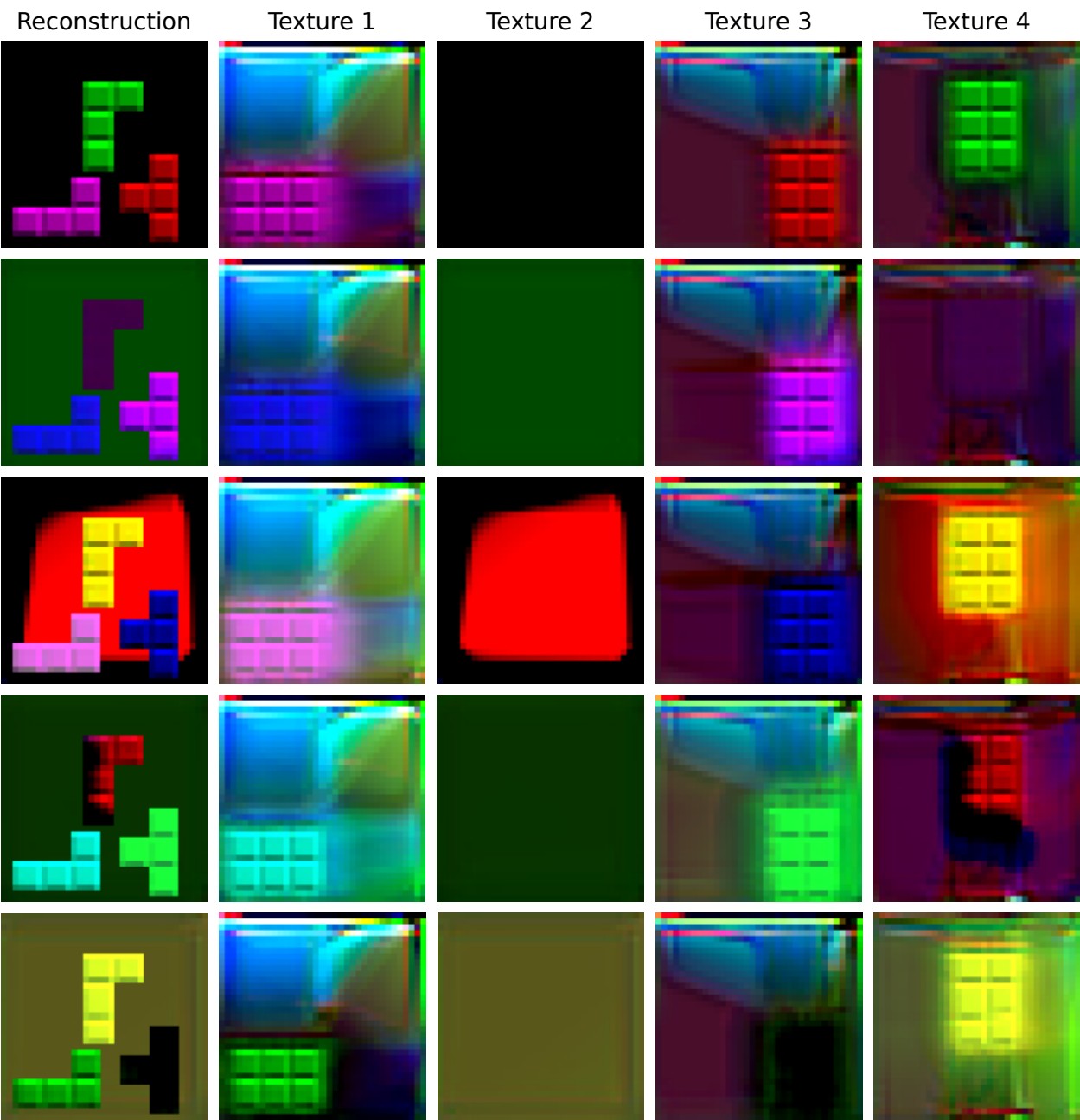

Figure 33: Texture generation on Tetrominoes. The first row shows final reconstruction and object textures obtained after encoding and decoding an input image. The second to fifth rows present final reconstructions and individual object textures after sampling new texture vectors.

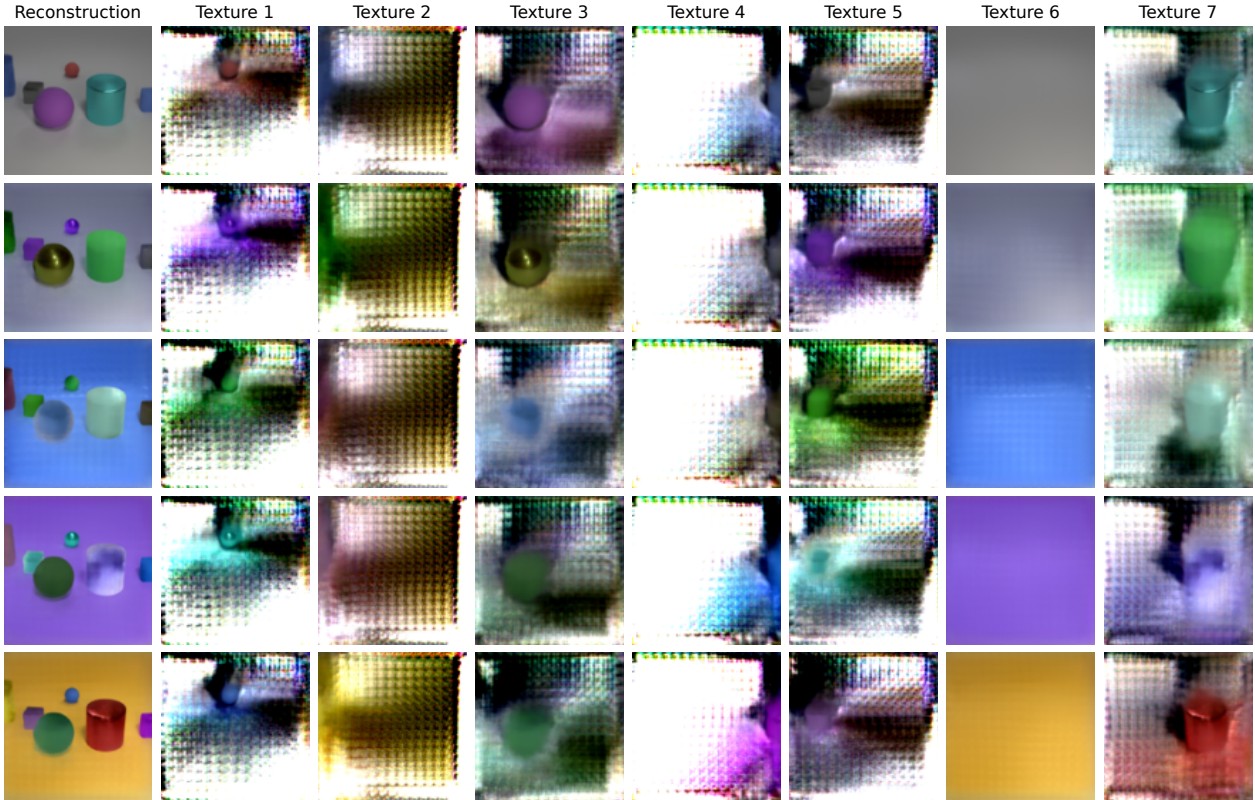

Figure 34: Texture generation on CLEVR6. The first row shows final reconstruction and object textures obtained after encoding and decoding an input image. The second to fifth rows present final reconstructions and individual object textures after sampling new texture vectors.

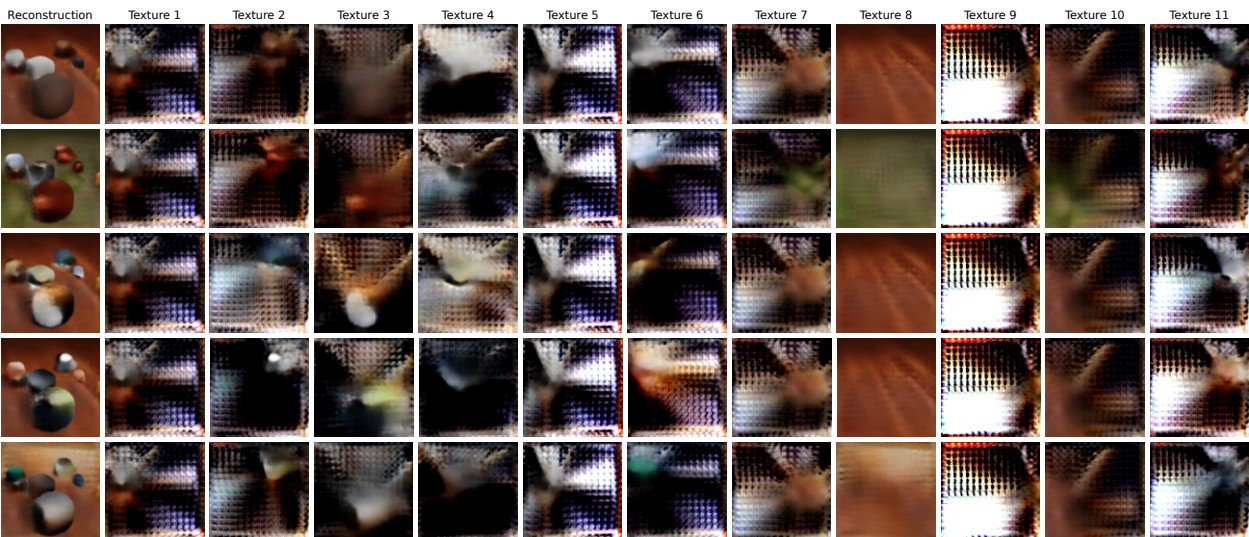

Figure 35: Texture generation on CLEVRTex. The first row shows final reconstruction and object textures obtained after encoding and decoding an input image. The second to fifth rows present final reconstructions and individual object textures after sampling new texture vectors.

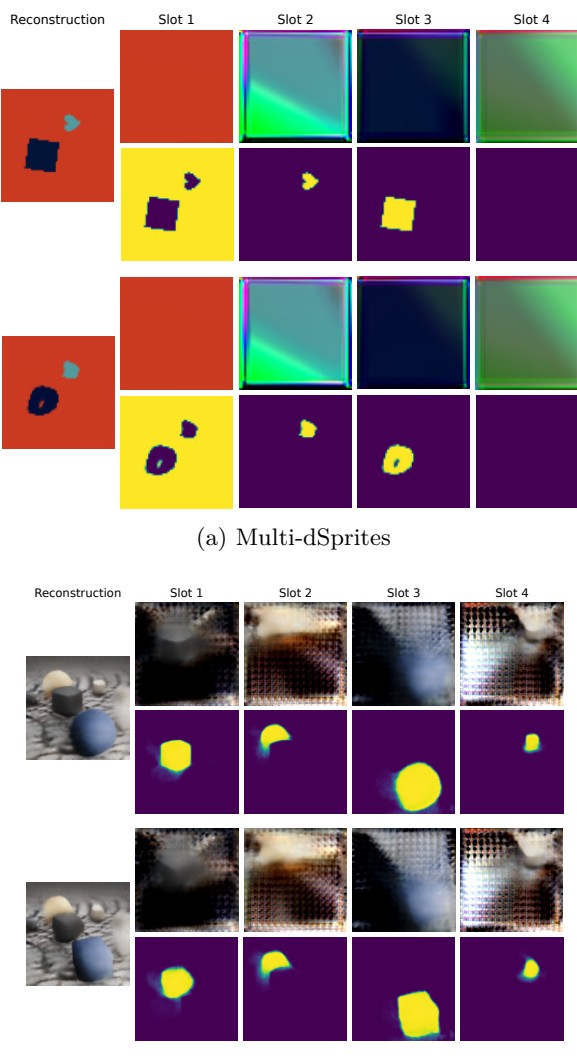

(a) Multi-dSprites

(b) CLEVRTex

Figure 36: Shape generation results on Multi-dSprites and CLEVRTex. In each of the 2 images, the first pair of rows shows final reconstruction (middle left), object textures (first row), and shapes (second row) obtained after encoding and decoding an input image. The second pair of rows presents final reconstructions (middle left) and individual object textures (first row) and shapes (second row) after sampling new shape vectors.

