# OpenReview forum: "Explicitly Disentangled Representations in Object-Centric Learning"
_TMLR — Accepted by TMLR_

### Review · Reviewer_kf8a · 2024-09-03

**Summary Of Contributions:**

This paper introduced an algorithm that builds on top of (invariant) Slot attention to explicitly disentangle the learned representation of shape and texture of input images. The algorithm works by using two encoder-decoder architecture by encoding object mask and object texture information in an object-centric manner. Using the learned representations, the algorithm was able to provide interpretability of the input image and was able to reconstruct the image using the learned representations.

**Audience:**

Yes

**Claims And Evidence:**

Yes

**Requested Changes:**

At this stage of the paper, I'd recommend authors to at least address some of my questions in weakness sections before future revisions

**Strengths And Weaknesses:**

Strengths
- The paper introduced backgrounds and related works in a decent manner. I personally find them to be useful references for those who are not familiar with the context
- - The algorithm is build on top of the invariant slot attention module and explicitly disentangle color and shape. This is a decent extension of the ISA algorithm.
- The settings of the algorithm makes sense on the high-level, and experimental results on some public benchmarks did demonstrate some effectiveness of the approach on disentangling texture and shape.

Weaknesses:
- Although the insight of the paper is sensible, I find the approach is only tested on synthetic and less complex datasets. The approach did demonstrate a proof-of-concept of the effectiveness of DISA algorithm, but I wonder if authors could discuss more on the what it might influence future algorithm designs, or real-world applications by demonstrating on more complex datasets.
- In most of the learned textures in the CLEVR dataset, there are heavy checkerboard artifacts. I wonder if there is an issue of the implementation of the algorithm -- The effect is not showing up on Tetrominoes datasets
- The experiment demonstrated the effect to reconstruct textures. Since the shape is also disentangled, I wonder if the algorithm could also reconstruct shapes?

---

> ### Author Response · Authors · 2024-10-28
> **Response to Reviewer kf8a**
>
> Dear Reviewer kf8a,
>
> Thank you for reviewing our work and for providing valuable concerns.
>
> - Regarding the first point, we recognize that experiments on more complex real-world datasets would contribute to further validating the practical effectiveness of DISA. Unfortunately, constraints on time and computational resources influenced our decision to limit testing on more complex datasets than CLEVRTex, especially as more complex and computationally expensive architectures should also be utilized. In our opinion, it would be even more interesting for future work to explore the scalability of DISA to more complex datasets when more powerful filters are identified and employed.
> - For question two, it is difficult to state exactly why certain artifacts emerge in the region unattended by a slot when reconstructing its texture. However, in our opinion, even if the reconstructions in those regions are not directly optimized, they are not random and tend to have some structure due to a combination of the model architecture, the training process, and the data at hand. This could explain why Tetrominoes, Multi-dSrites (Appendix F.3), and CLEVR all present different artifacts (while CLEVRTex has similar ones to CLEVR). We are also confident that there is no issue on that side of the implementation, but since we could be wrong, we welcome everyone to check the code included in the supplementary materials.
> - The last question is indeed interesting. The model is very good at reconstructing shapes present in the training data. The reconstruction is performed by the mask decoder (bottom right, Figure 1). You can find the associated quantitative results in Table 1 and qualitative ones in the middle row of Figure 3, other than many more in section F.3 of the Appendix. However, generating new shapes is a far more difficult task, and we therefore included Figure 35, Appendix F.4, to address how the model behaves in this setting. Consider that, as written in Appendix F.4: "... this task is quite difficult: first, the number of unique shapes is strongly limited in these synthetic datasets; second, without changing the scale of the reference frame, the generated shape is restricted to one that fits the original frame.". We would also like to highlight that, in Figure 14 (Appendix F.2), we experiment with scene reconstruction with varying object positions and sizes, while keeping shapes and textures fixed, to show the position and scale disentanglement inherited from the ISA mechanism.
>
> We hope to have exhaustively addressed all your concerns.

---

### Review · Reviewer_sYNF · 2024-10-14

**Summary Of Contributions:**

The paper introduces a novel approach named Disentangled Slot Attention (DISA) to enhance object-centric learning through explicit disentanglement of object attributes, focusing primarily on texture and shape components. Building upon Invariant Slot Attention (ISA), the proposed architecture ensures non-overlapping subsets of latent space dimensions for these features, thus improving interpretability and performance. DISA demonstrates competitive performance in object discovery and reconstruction across several synthetic datasets, along with strong compositional and generative capabilities.

**Audience:**

Yes

**Claims And Evidence:**

Yes

**Requested Changes:**

- Though not required, I was hoping to see an ablation study on how the model by using stronger/diffferent filters would perform on the given datasets.

- I think adding some summarization of the findings in figures, especially fig 2, would make those a lot clearer.

- Related work might need to be tighten.

**Strengths And Weaknesses:**

Strength:

- DISA offers a unique extension to ISA by explicitly disentangling shape and texture features, enhancing both the interpretability and effectiveness of object-centric representations.
- The paper evaluates DISA on several challenging benchmarks, such as CLEVR6, and CLEVRTex, and shows improved or competitive performance in terms of reconstruction quality and object discovery.
- The contribution of this study is clear

Weakness:

- Some datasets, such as CLEVR6 and CLEVRTex, show leakage of information between shape and texture components, indicating the disentanglement is not always perfect.
- The use of solber filter may have difficulties for more complex data, as discussed in the paper.

---

> ### Author Response · Authors · 2024-10-28
> **Response to Reviewer sYNF**
>
> Dear Reviewer sYNF,
>
> Thank you for your review and for raising fair concerns.
>
> - Concerning the first requested change, we agree that experimenting with additional and more powerful filters would have improved the paper. At first, we thought about including a learned filter based on an information bottleneck alongside the Sobel. However, we ended up deciding to leave it for future work, where a wider set of filters could be identified and studied more thoroughly, especially on more complex datasets.
> - Regarding the suggestion about summarizing the findings in the figures, we again agree and have revised the paper accordingly (Figures 2, 3, and 4).
> - For the last point, we understand your concern. However, we believe that is important to provide the readers with a more complete overview of the context in which the paper is positioned, especially for those less familiar with this area of research.
>
> We hope we were exhaustive with our answers.

---

### Review · Reviewer_gmpn · 2024-10-16

**Summary Of Contributions:**

The paper introduce a method for disentangled representation learning from an object-centric perspective. In the proposed framework, two prior feature extractor and encoder is used to encode the input image into two seperate embeddings that contain the texture and shape information of the original information. To aviod the positioning and scaling issues between two embedding spaces, the authors further use slot attension to align the texture and shape embeddings. Extensive experiments have been conducted to shown the effectiveness of the method in finding the texture and shape generative factors, hence validating the disentanglement of features using the method.

**Audience:**

Yes

**Claims And Evidence:**

Yes

**Requested Changes:**

It is suggested that the authors address the insufficientness of the the experiments sections.

**Strengths And Weaknesses:**

Strengths:
1. The paper introduce a novel method of object-centric disentangled representaion learning, which is interesting. The idea offers an intuitive way of achieving disentanglement and is convincing to the audience.
2. The writing of the paper is clear and easy to follow.

Weaknesses:
1. The quantitative comparisons are only conducted with variations of the proposed methods, it is recommanded to include comparision with existing works to show the effectiveness of the proposed methods.
2. The method requires strong prior knowledges about the data, for example, the edges from sobel filter or the texture. This might limits the potential usage and impact of the method as the generating factors are pre-defined and restricted.

---

> ### Author Response · Authors · 2024-10-29
> **Response to Reviewer gmpn**
>
> Dear Reviewer gmpn,
>
> Thank you for reviewing our work and for providing helpful feedback.
>
> - Regarding the first point, as written in Section 5, our work seeks to achieve the desired disentanglement within the latent space of DISA rather than focusing on state-of-the-art results in unsupervised object discovery and reconstruction quality. For this reason, the aim of the experiments is to show that it is possible to induce the desired property in a Slot Attention (SA) type of model without compromising its performance. To demonstrate that the proposed extension does not degrade the object discovery and reconstruction performances of its baseline, only ARI and MSE of DISA (proposed) and (I)SA (baselines) must be compared. Concerning the disentanglement analysis, we did not find it relevant to our goal to include a comparison with other methods.
> - For the second point, while it is true that knowledge about the data could be valuable in defining a priori the optimal number of latent space dimensions related to texture and shape, we believe that it is not a limiting factor for the potential usage and impact of our method. The reason is that, when we scale the complexity of the data (i.e., more variable and detailed textures, more shapes, etc.) and it becomes impossible to have that knowledge a priori, we can always overestimate it. Doing so does not seem to be a limiting factor from our experiments, since we already overestimated those numbers (32 dimensions for texture and 32 for shape) without compromising the performances. Moreover, further experiments can be done to subsequently reduce the dimensionality until possible. The prior knowledge about the data can also be useful for an a priori intuition on whether or not the chosen filter can sufficiently filter out the texture information. However, we think that if in future work a sufficiently good filter is found for complex real-world data, then that filter can be used on all less complex data (similarly in concept to the overestimation of the dimensionalities) without compromising performances. This would make, with time, less and less important the a priori knowledge about the data when applying our proposed method.
>
> We hope to have provided sufficiently exhaustive answers.

---

### Decision · Action_Editor_JRjK · 2024-12-17

**Recommendation:** Accept as is

**Comment:**

The reviewers were split in opinion on this paper. Arguments in favor of the paper included thorough experimental validation of the approach and the presence of interesting ideas regarding disentanglement. Arguments against included skepticism about whether the proposed architecture can generalize to more complex datasets and uncertainty about the presence of artifacts in the results and the effect of the image filter. These points are all valid and well-received. However, it should be noted that the acceptance criteria for TMLR are twofold: (1) the claims should supported by clear evidence, and (2) at least some individuals in TMLR's audience should be interested in the findings. The answer to these questions are both "yes", as discussed above. Nevertheless, the authors are strongly encouraged to incorporate relevant feedback from the reviewers into the revision, as the suggested changes would strengthen the paper. In addition, the authors are encouraged to move important material from the appendix into the main body of the paper according to their judgment. Examples may include (but are not limited to) the related work in Appendix C, the visualization of property prediction in Figure 9, and selected qualitative results from Appendix F (Figures 14-35), as appropriate.

**Audience:**

There would be at least some individuals interested in the findings of this paper. Researchers interested in compositional generation of shape and textures would find a direct application of the proposed architecture in their work. At a more abstract level, researchers interested in disentanglement in general may find inspiration from the approach taking by DISA: that of an information removing procedure (in this case, a Sobel filter) followed by separate encoders.

**Claims And Evidence:**

Three main claims are made in this paper. The first is the claim of a novel object-centric representation learning architecture that has a bias towards disentangling shape and texture. Support for the novelty of the proposed method, called DISA, is presented in the related work (Section 3 and Appendix C). The second claim is that DISA improves baseline performance in most cases while simultaneously achieving disentanglement of shape and texture. Support for this claim is presented in both Section 5.1 "Object Discovery and Reconstruction" and Section 5.2 "Disentanglement Analysis". The former shows an improvement in both MSE and ARI relative to the Slot Attention and Invariant Slot Attention models upon which DISA is based. The latter claim is tested through property prediction and inverse property prediction results, which show that information about shape is largely confined to the shape representation while information about texture is largely confined to the texture representation. The third and final claim is that DISA supports texture transfer between objects and the generation of novel texture. Support for this last claim is provided in Section 5.3, which shows that novel textures can be sampled from a Gaussian distribution and texture transfer can occur by permuting the object texture representations. In summary, the claims are appropriately scoped with sufficient evidence provided to support each.

---

> ### Author Response · Authors · 2025-01-01
> **Official Comment by Authors**
>
> We express our sincere gratitude to the AE and all reviewers for their time and valuable feedback.
>
> We revised the paper according to the suggestions and submitted the camera-ready version.